# Calibrated Value-Aware Model Learning with Probabilistic Environment Models

**Claas Voelcker** [1 2]   **Anastasiia Pedan** [3 1 2]   **Arash Ahmadian** [4]   **Romina Abachi** [5]   **Igor Gilitschenski** [1 2]
**Amir-massoud Farahmand** [6 7 1]

## Abstract

The idea of value-aware model learning, that models should produce accurate value estimates, has gained prominence in model-based reinforcement learning. The MuZero loss, which penalizes a model's value function prediction compared to the ground-truth value function, has been utilized in several prominent empirical works in the literature. However, theoretical investigation into its strengths and weaknesses is limited. In this paper, we analyze the family of value-aware model learning losses, which includes the popular MuZero loss. We show that these losses, as normally used, are uncalibrated surrogate losses, which means that they do not always recover the correct model and value function. Building on this insight, we propose corrections to solve this issue. Furthermore, we investigate the interplay between the loss calibration, latent model architectures, and auxiliary losses that are commonly employed when training MuZero-style agents. We show that while deterministic models can be sufficient to predict accurate values, learning calibrated stochastic models is still advantageous.

## 1. Introduction

In model-based reinforcement learning, an agent collects information in an environment and uses it to learn a model of the world. This model is used to improve value estimation and policy learning (Sutton, 1990; Deisenroth & Rasmussen, 2011; Hafner et al., 2020; Schrittwieser et al., 2020). However, as environment complexity increases, learning a model becomes more and more challenging. This leads to model

errors which propagate to value function learning (Schneider, 1997; Kearns & Singh, 2002; Talvitie, 2017; Lambert et al., 2020). In such cases, deciding what aspects of the environment to model is crucial.

The paradigm of *value-aware model learning* (VAML) (Farahmand et al., 2017) and *value-equivalence* (Grimm et al., 2020; 2021) addresses this by training models that lead to accurate value estimation. Prominent value-aware model learning approaches are MuZero (Schrittwieser et al., 2020) and IterVAML (Farahmand, 2018). The MuZero loss has been shown to perform well in discrete (Schrittwieser et al., 2020; Ye et al., 2021) and continuous control tasks (Hansen et al., 2022; 2024), but has received little theoretical investigation. On the other hand, IterVAML is a theoretically motivated algorithm but not commonly used in empirical work. We show that MuZero and IterVAML can be unified in a family of losses, which we term $(m, b)$-Value-Aware Model Losses ($(m, b)$-VAML). The name stresses the two core hyperparameters: the model rollout steps, $m$, and steps used to estimate the bootstrapped value function target, $b$.

The $(m, b)$-VAML losses are used as surrogate losses in place of other value- or model learning losses. Therefore, it is important to ask whether they are calibrated (Steinwart & Christmann, 2008). A calibrated loss does not lead to suboptimal minima when the function class includes optimal functions for the original target loss.

**Research question:** This paper has two parts, each with a theoretical and empirical section. We answer two questions about the $(m, b)$-VAML family: (a) What variants of the $(m, b)$-VAML losses are well-calibrated to recover correct models and value functions? (b) Do we observe problems with uncalibrated losses when using standard architectures, especially deterministic latent-space models?

**Contributions:** As our main theoretical contribution, we mathematically analyze the family of $(m, b)$-VAML algorithms. We prove that all sampled-based loss variants from this family are uncalibrated when used with a stochastic environment model. Minimizing the losses with regard to data samples will result in value functions and models with lower variance than the correct ground-truth functions, even

---

[1]Department of Computer Science, University of Toronto, Canada [2]Vector Institute, Toronto, Canada [3]Igor Sikorsky Kyiv Polytechnic Institute, Kyiv, Ukraine [4]Cohere, Toronto, Canada [5]Ubisoft, Montreal, Canada [6]Polytechnique Montreal, Canada [7]MILA, Montreal, Canada. Correspondence to: Claas Voelcker <cvoelcker@cs.toronto.edu>.

*Proceedings of the 42$^{st}$ International Conference on Machine Learning*, Vancouver, Canada. PMLR 267, 2025. Copyright 2025 by the author(s).

if the dataset adequately covers the state-action space. To counter this issue, we derive a novel loss variant.

In the second part, we address issues arising from the way current algorithms in the $(m, b)$-VAML family are commonly implemented. We prove that a stochastic model class is not necessary to learn a single-step decision-equivalent model in stochastic environments. This validates the practice of primarily using deterministic models in empirical work (Oh et al., 2017; Schrittwieser et al., 2020; Hansen et al., 2022). Empirically we find that using stochastic models can still lead to improved performance, although this is environment-dependent.

## 2. Background

**Reinforcement Learning:** We consider a standard Markov Decision Process (MDP) (Puterman, 1994) $(\mathcal{X}, \mathcal{A}, \mathcal{P}, r, \gamma)$, with state space $\mathcal{X}$, action space $\mathcal{A}$, transition kernel $\mathcal{P}(x'|x, a)$, reward function $r : \mathcal{X} \times \mathcal{A} \to \mathbb{R}$, and discount factor $\gamma \in [0, 1)$. A policy $\pi(a|s)$ maps a state to a distribution over actions. The goal of a reinforcement learning agent is to find a policy maximizing the expected discounted infinite horizon reward, i.e., $\max_\pi \mathbb{E}_{\pi, \mathcal{P}} \left[ \sum_{t=0}^\infty \gamma^t r(x_t, a_t) \, | \, x_0 \right]$.

The value function is defined as the expected return of the policy $\pi$: $V^\pi(x) = \mathbb{E}_{\pi, \mathcal{P}} \left[ \sum_{t \geq 0} \gamma^t r(x_t, a_t) | x_0 = x \right]$. The policy-conditioned transition kernel is $\mathcal{P}^\pi(x'|x) = \int \mathcal{P}(x'|x, a) \pi(a|x) \mathrm{d}a$. The value is the unique fixed point of the Bellman operator

$$[\mathcal{T}_{\mathcal{P}^\pi} V](x) = \mathbb{E}_\pi \left[ r(x_0, a_0) + \gamma \mathbb{E}_{\mathcal{P}^\pi} [V(x_1)] | x_0 = x \right].$$

This operator can be extended to a multi-step version as

$$[\mathcal{T}_{\mathcal{P}^\pi}^b V](x) = \mathbb{E}_{\pi, \mathcal{P}} \left[ \sum_{n=0}^{b-1} \gamma^n r(x_n, a_n) + \gamma^b V(x_b) \Big| x_0 = x \right].$$

We also define $[\mathcal{T}_{\mathcal{P}^\pi}^0 V](x) = V(x)$.

**Model-based RL:** An *environment model* is a function that approximates the transition kernel.[1] Learned models are used to augment RL algorithms in several ways. For a comprehensive survey, refer to Moerland et al. (2023). In this paper, we focus on value learning with model data, which is commonly referred to as Dyna (Sutton, 1990).

We use $\hat{p}$ to refer to a stochastic model, and $\hat{f}$ for deterministic models. When a model is used to predict the observation $x'$ from $x, a$ (such as the model used in MBPO (Janner et al., 2019) we call it an *observation-space models*. Alternatively, *latent-space models* of the form $\hat{p}(z'|z, a)$ are used,

where $z \in \mathcal{Z}$ is a representation of a state $x \in \mathcal{X}$ given by $\varphi : \mathcal{X} \to \mathcal{Z}$. Observation-space models predict next states in the representation of the environment, while latent-space models reconstruct learned features of these states.

The notation $x^{(n)}$ refers to the $n$-th step in a rollout starting in state $x$ in the environment. We will use $\hat{x}^{(n)}$ to refer to samples from the $n$-th step model prediction and write $\mathbb{E}_{\hat{p}^m} [\cdot]$ as a shorthand for $\mathbb{E}_{x^{(m)} \sim \hat{p}^{(m)}(\cdot|x)} [\cdot]$.

## 3. The Value-Aware Model Learning framework

The losses of the decision-aware learning framework share the goal of finding models that provide good value function estimates. Instead of simply learning a model using maximum likelihood estimation, the losses are based on differences in value prediction.

### 3.1. Iterative Value Aware Model Learning

Iterative Value Aware Model Learning (IterVAML) (Farahmand, 2018) computes the difference between the expected value under the model and samples in the environment

$$
\begin{aligned}
&\mathcal{L}_{\mathrm{IterVAML},m}^{\mathcal{P}^\pi} (\hat{p}, V|x) \\
&= \left| \mathbb{E}_{\hat{p}^m} \left[ V \left( \hat{x}^{(m)} \right) \right] - \left[ \mathbb{E}_{\mathcal{P}^\pi} \left[ V \left( x^{(m)} \right) \right] \right]_{\mathrm{sg}} \right|^2 \quad (1) \\
&\approx \left| \frac{1}{K} \sum_{k=1}^K \left[ V \left( \hat{x}_k^{(m)} \right) \right] - \left[ V \left( x^{(m)} \right) \right]_{\mathrm{sg}} \right|^2.
\end{aligned}
$$

We will use $\mathcal{L}_{\mathrm{IterVAML},m}$ to refer to the expectation-based version, and $\hat{\mathcal{L}}_{\mathrm{IterVAML},m}^k$ to refer to the sampling-based version. $[\cdot]_{\mathrm{sg}}$ denotes the stop-gradient operation. To reduce notational complexity, we drop the action dependence in all following propositions; all results hold without loss of generality for the action-conditioned case as well. The expectation-based version of the IterVAML loss has an important relationship to the error of computing the model's Bellman operator compared to the true environments Bellman operator.

**Proposition 1.** *Farahmand (2018) Let $\mathcal{P}^\pi$ be the policy-conditioned transition kernel of an MDP and let $V : \mathcal{X} \to \mathbb{R}$ be a function. Let $\hat{p}(\cdot|x)$ be a model so that $\mathcal{L}_{\mathrm{IterVAML},1}^{\mathcal{P}^\pi} (\hat{p}, V|x) = 0$. Then $[\mathcal{T}_{\mathcal{P}^\pi} V] (x) = [\mathcal{T}_{\hat{p}} V] (x)$.*

We give a short proof in Subsection A.1

Intuitively, a model achieving 0 loss can be used instead of the ground truth environment when computing the Bellman operator. The set of models that achieve 0 IterVAML error is equal to the set of value-equivalent models for the value estimate $\hat{V}$ (Grimm et al., 2021).

---

[1] In this paper, we will generally use the term *model* to refer to an environment model, not a neural network, to keep consistent with the reinforcement learning nomenclature.

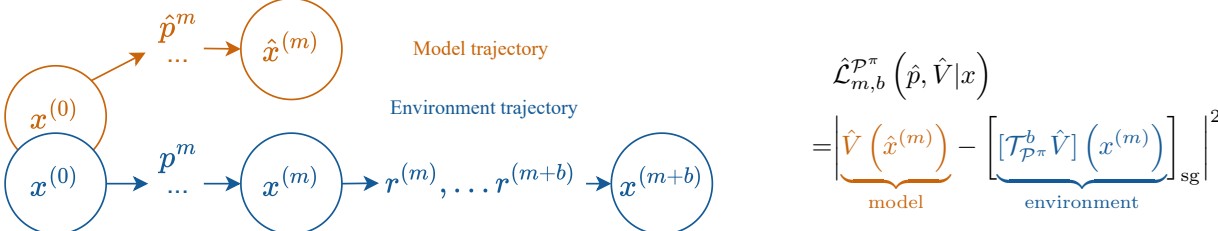

*Figure 1.* Sketch of $(m, b)$-VAML. The loss is computed from an $m$-step model and a $(m + b)$-step environment trajectory. It is the difference between the estimated value of the $m$-th model state, and the $b$-step Bellman operator starting from the $m$-th environment state.

## 3.2. The (m,b)-VAML Family

The MuZero loss (Schrittwieser et al., 2020) was introduced to unify the value function and model learning components of an MBRL algorithm. It can be interpreted as a variation of the IterVAML loss that uses a single sample estimate of the Bellman Operator and a bootstrapped target estimate. To unify the losses, we present them in a single equation with two important hyperparameters: $m$, the number of steps in the trajectory that the loss is computed over, and $b$, the number of steps for the multi-step Bellman operator. We refer to this unified family of losses as $(m, b)$-Value Aware Model Losses (VAML)

$$
\hat{\mathcal{L}}_{m,b}^{\mathcal{P}^\pi} \left( \hat{p}, \hat{V} | V_{\text{tar}}, x \right)
$$
$$
= \mathbb{E}\left[ \left| \hat{V}\left( \hat{x}^{(m)} \right) - \left[ \left[ \mathcal{T}_{\mathcal{P}^\pi}^b V_{\text{tar}} \right]\left( x^{(m)} \right) \right]_{\text{sg}} \right|^2 \Big| x \right] \quad (2)
$$

where $V_{\text{tar}}$ is a target network (Mnih et al., 2013) and $x^{(m)}$ and $\hat{x}^{(m)}$ are sampled independently from $\mathcal{P}^\pi$ and $\hat{p}$ respectively. Note that samples from the real environment are used to approximate $[\mathcal{T}_{\mathcal{P}^\pi}^b V_{\text{tar}}]$. The loss function and the relation of the model and environments rollout are visualized in Figure 1.

Several works use variations of this loss: The original MuZero algorithm (Schrittwieser et al., 2020) and follow-up work (Ye et al., 2021; Antonoglou et al., 2022) use $m \geq 1$ and $b \geq 1$. In continuous control, $m \geq 1, b = 1$ has been used in the TD-MPC line of work (Hansen et al., 2022; 2024). When using only a single sample $k = 1$, the sample-based IterVAML loss is equal to $\hat{\mathcal{L}}_{m,0}^{\mathcal{P}^\pi}$. Note that the $(m, b)$-loss can easily be extended to a $k$ sample variant analogous to the IterVAML loss, which we will use later. We dropped the summation over $k$ samples here to simplify the (already dense) notation. Finally, regular model-free TD learning corresponds to $m = 0, b \geq 1$.

# 4. Analysis of decision-aware losses in stochastic environments

The goal of learning in MBRL is to recover an (approximately) optimal model and to learn a correct value function. As we have shown, minimizing the IterVAML loss perfectly results in a model that leads to a correct Bellman Operator. However, in practice, the inner expectation of the IterVAML loss has to be replaced by a sampling-based approximation. In addition, $(m, b)$-VAML is used in the MuZero algorithm to update the value function directly in addition to the model. We show when this leads to learning correct models and value functions asymptotically. An overview of our conclusions can be found in Table 1.

## 4.1. Calibration of surrogate loss functions

Formally, we ask whether the surrogate $(m, b)$-VAML is *calibrated*. Intuitively, a calibrated surrogate loss does not select a suboptimal function for the target loss. Formally, we require that the surrogate loss is not perfectly minimized by a function that does not perfectly minimize the target loss. Therefore, we call losses that do not have this property *minimum-uncalibrated* losses, or simply uncalibrated outside of formal statements. We borrow the concept from Steinwart & Christmann (2008), however, we use a slightly more restricted definition of *uncalibrated* here. This means that all losses which we consider uncalibrated are also uncalibrated in the definition given in Steinwart & Christmann (2008), but not vice versa.

**Definition 1** (Minimum-uncalibrated surrogate losses). *Let* $\mathcal{L}_{\text{tar}}(f, x, x^{(m)})$ *be a loss function defined over samples from an MDP. Let* $\hat{\mathcal{L}}_{\text{sur}}$ *be a surrogate function for the loss. Let* $\mathcal{F}^*$ *be a set of (perfect) minima of* $\mathcal{L}_{\text{tar}}$ *so that* $\mathcal{L}_{\text{tar}}(f^*, x, x^{(m)}) = 0$ *for all* $f^* \in \mathcal{F}^*$. *A surrogate function is minimum-uncalibrated for the target loss if there exists an MDP, a function class* $\mathcal{F}$ *with* $\mathcal{F}^* \subseteq \mathcal{F}$, *and a state* $x$ *so that*

$$
\arg \min_{f \in \mathcal{F}} \mathbb{E}_{x^{(m)} \sim \mathcal{P}^\pi(\cdot | x)} \left[ \mathcal{L}_{\text{sur}}(f, x, x^{(m)}) \right] \not\subseteq \mathcal{F}^*.
$$

An example of minimum-uncalibrated-ness is the *double sampling issue* in Bellman Residual Minimization (BRM). While the target loss is minimized by the ground truth value function, BRM actually chooses a function that minimizes the Bellman error and and additional variance term.

We will show that some issues introduced by naively using $(m, b)$-VAML can be fixed with a tractable modification to the loss. We call such cases *resolvable*.

### 4.2. Model learning bias with stochastic models

Most prior work uses $(m, b)$-VAML with deterministic models. Exceptions are Voelcker et al. (2022) and Antonoglou et al. (2022), but neither changes the loss functions to account for the model parametrization. For a stochastic model class, $(m, 0)$-VAML is uncalibrated for the target loss $\mathcal{L}_{\text{IterVAML}}$.

We begin with analyzing the loss for $m \geq 1$ and $b = 0$. For simplicity we assume that $V_{\text{tar}} = \hat{V}$.

**Proposition 2.** *Let $\hat{\mathcal{L}}_{m \geq 1, 0}^{\mathcal{P}^\pi}(\hat{p}, V|x, x^{(m)})$ be the surrogate loss for $\mathcal{L}_{\text{IterVAML},m}^{\mathcal{P}^\pi}(\hat{p}, V|x)$. Let $\mathcal{P}^*$ be the set of all distributions $p$ for which $\mathbb{E}_p\left[V(\hat{x}^{(m)})\right] = \mathbb{E}_{\mathcal{P}^\pi}\left[V(x^{(m)})\right]$. There exist an MDP and a class of distributions $\mathbb{P}$ with $\mathcal{P}^\pi \in \mathcal{P}^* \subseteq \mathbb{P}$ so that*

$$\underset{\hat{p} \in \mathbb{P}}{\text{Arg min}} \, \mathbb{E}_{\hat{p}}\left[\hat{\mathcal{L}}_{m \geq 1, 0}^{\mathcal{P}^\pi}(\hat{p}, V|x, x^{(m)})\right] \not\subseteq \mathcal{P}^*.$$

*Therefore, $\hat{\mathcal{L}}_{\text{model}}$ is minimum-uncalibrated.*

Proofs for this section can be found in subsubsection A.2.2. When using samples from a stochastic model to compute the loss function in Equation 2, we are left with a variance error term that is closely related to the double-sampling problem

$$\mathbb{E}_{\hat{p}}[\hat{\mathcal{L}}_{1,0}^{\mathcal{P}^\pi}(\hat{p}, V|x)]$$
$$= \mathcal{L}_{\text{IterVAML},1}^{\mathcal{P}^\pi}(\hat{p}, V|x, x') + \text{Var}_{\hat{p}}(V(\hat{x})). \quad (3)$$

In the classic double-sampling problem, we cannot correct the variance term as we do not have oracle access to the environment. Here the issue is our model, and we can generate multiple samples from it. Therefore, we can estimate this variance term from samples and correct the loss. This correction is reminiscent of Antos et al. (2008) but is simpler to obtain as we only need model samples.

To obtain the correction, we define $\hat{\mu}_{\hat{p}}^{m,k} = \frac{1}{k}\sum_{i=1}^k V(\hat{x}_i^{(m)})$, the empirical estimator of the expected return in Equation 1. The variance of this estimator can be estimated as $\widehat{\text{Var}}_{\hat{p}}^{m,k} = \frac{1}{k}\sum_{i=1}^k (V(\hat{x}_i^{(m)}) - \mu_{\hat{p}}^{m,k})^2$. With this we can define a new loss which can be computed with at least two samples from the model

$$\hat{\mathcal{L}}_{\text{CVAML},m}^k = \hat{\mathcal{L}}_{\text{IterVAML},m}^k - \widehat{\text{Var}}_{\hat{p}}^{i,k}.$$

We refer to the loss as *Calibrated VAML* (CVAML).

**Proposition 3.** *The variance-corrected loss $\hat{\mathcal{L}}_{\text{CVAML},m}^k(\hat{P}, V|x, x^{(m)})$ is a calibrated surrogate loss for $\mathcal{L}_{\text{IterVAML},m}^{\mathcal{P}^\pi}(\hat{P}, V|x, x^{(m)})$.*

Analogous to the IterVAML case, the sample-based $(m, b \geq 1)$-VAML loss is an uncalibrated loss for learning a model.

### 4.3. Value learning bias in stochastic models

We now show that this issue also affects the value function learning with the MuZero loss. The problem here lies in the use of the bootstrapped Bellman target together with a multi-step value rollout. In an MDP, the values of two states $x$ and $y$ are not guaranteed to be equal (or even particularly close) just because they share an ancestor state, unless we make assumptions about the variance of the value function over successor states. However, the MuZero loss still minimizes the difference in value functions between these two states.

We show that the MuZero loss therefore is not guaranteed to recover the correct value function, even when we have a perfect (stochastic) model. To formalize the issue, we compare the solution found with $(m, b)$-VAML with the regular TD loss function. Note that for this loss, we assume that $x_{(m)}$ is a fixed sample from the environment, *not* a random variable like $x^{(m)}$, with $x_{(m+1)}$ being its successor state in the ground truth environment. This distinction is important, as it is exactly what leads to the bias in the $(m, b \geq 1)$ loss.

$$\mathcal{L}_{\text{TD}}(\hat{V}|V_{\text{tar}}, x^{(m)}, x^{(m+1)}, r^{(m)})$$
$$= \left(\hat{V}(x^{(m)}) - \left[r^{(m)} + \gamma V_{\text{tar}}(x^{(m+1)})\right]\right)^2. \quad (4)$$

To drive home that this problem is independent of the model error, we show that the $(m, b \geq 1)$-VAML loss is uncalibrated even if we substitute the ground truth environment for the learned model, and focus solely on the value learning component.

**Proposition 4.** *Let $\mathcal{L}_{\text{TD}}(V|V_{\text{tar}}, x^{(m)})$ be the target loss, and let $\hat{\mathcal{L}}_{m,1}^{\mathcal{P}^\pi}(\mathcal{P}^\pi, V|V_{\text{tar}}, x, x^{(m),r^{(m)}})$ be the surrogate loss. There exists a set of functions $\mathcal{V}$ for any $V_{\text{tar}}$ that is not a constant function, for which two conditions hold:*

1. *The set is complete, meaning that $[\mathcal{T}_{\mathcal{P}^\pi} V_{\text{tar}}] \in \mathcal{V}$ for some target function $V_{\text{tar}}$,*

2. *and,*

$$\underset{\hat{V} \in \mathcal{V}}{\text{Arg min}} \, \mathbb{E}_{\mathcal{P}^\pi}\left[\hat{\mathcal{L}}_{m,1}^{\mathcal{P}^\pi}(\mathcal{P}^\pi, \hat{V}|V_{\text{tar}}, x, x^{(m)})\right] \not\subseteq [\mathcal{T}_{\mathcal{P}^\pi} V_{\text{tar}}].$$

*Therefore, $\hat{\mathcal{L}}_{m,1}^{\mathcal{P}^\pi}$ is minimum-uncalibrated.*

| Model | Env. | IterVAML $(m \geq 1, b = 0)$ | MuZero $(m \geq 1, b \geq 1)$ | Model-free TD $(m = 0, b \geq 1)$ |
|---|---|---|---|---|
| Det. model, | det. env | calibrated | calibrated | calibrated |
| Stoch. model, | det. env | uncalibrated (resolvable) | uncalibrated (for VF updates) | calibrated |
| Det. model, | stoch. env | calibrated | calibrated | calibrated |
| Stoch. model, | stoch. env | uncalibrated (resolvable) | uncalibrated (for VF updates) | calibrated |
| Can update model | | ✓ | ✓ | X |
| Can update VF | | X | ✓ (for det. model) | ✓ |

*Table 1.* Comparison of the major design choices in the $(m, b)$-VAML framework. IterVAML and MuZero can be used to update the model, while MuZero and Model-free TD learning can be used to update the value function. All model-based losses are uncalibrated when applied to stochastic model classes. In addition, MuZero suffers a bias when used for value function prediction which cannot be surmounted with an easy modification to the loss function.

The problem in the loss function again depends on the variance of the value function with regard to the model. Proofs for this section can be found in subsubsection A.2.3.

To overcome this issue, we can introduce another variance correction term, similar to $\hat{\mathcal{L}}_{\text{CVAML}}$. This retains the advantage that the MuZero loss is used for both model and value updates. However, as we average over the value prediction from the model, we only guarantee that

$$\mathbb{E}_{\hat{p}} \left[ V(\hat{x}^{(m)}) \right] \approx \mathbb{E}_{\mathcal{P}^\pi} \left[ [\mathcal{T}_{\mathcal{P}^\pi}^b V_{\text{tar}}](x^{(m)}) \right], \quad (5)$$

not that the values for each state are correct. However, these are necessary for accurate planning or policy improvement. Therefore, it is still important to use a loss variant with $m = 0$ for learning accurate values for each state.

### 4.4. Discussion of theoretical results

While the problem resulting from the use of stochastic models is resolvable, the MuZero loss is still insufficient for learning *per state* value function with stochastic models. We observe issues with learning accurate value functions with the $(m \geq 1, b \geq 1)$-VAML loss even after correctly calibrating it. If the calibrated loss is used to update the model, and a standard model-free or model-based value estimate is used as the value function learning target, the correct value function can be learned.

> **Insight 1** (Calibrated losses). *To obtain a calibrated loss, we propose using $\hat{\mathcal{L}}_{\text{CVAML}}$ to update the model and a regular model-based or model-free TD loss to update the value function.*

## 5. Calibration impact on finite state MDPs

To test our findings, we run the $(m, b)$-VAML losses on small, finite state Garnet problems (Bhatnagar et al., 2007). We use the MuZero-style $(1, 1)$-VAML loss to update both model and value function, while the IterVAML-style $(1, 0)$-VAML loss is only used to train the model, and we use a

regular model-based TD loss to update the value function. For the baseline, we similarly use a model-based TD loss together with a KL-based loss.

We use $n$ to denote the size of the state space, $k$ for the number of successor states in the garnet, and $j$ for the rank of the model. Every problem is generated by sampling $k$ successor states for each state $x_i$, and parameterizing the transitions with weights $\omega_{i,j}$ sampled iid from a standard normal distribution for all successor states $x_j$. The transition distribution is given by $p(x_i'|x_l) = e^{\omega_{i,l}/\tau}/\sum_i e^{\omega_{i,l}/\tau}$. Varying the temperature $\tau$ we can interpolate between a deterministic transition and a uniform one.

The models are parameterized with two learnable matrices $\varphi \in \mathbb{R}^{j \times n}$ and $\psi \in \mathbb{R}^{j \times n}$ so that $\hat{p}_k(x_i|x_l) = \text{softmax}(\hat{\omega}_{i,l}) = \text{softmax}(\varphi_i^\top \psi_l)$. By varying $j$ we create a low-rank constraint. As Farahmand et al. (2017) shows, $(m, b)$-VAML should be used when the model has insufficient capacity to represent the environment. As a baseline, we use the model $\hat{p}_{\text{KL},k} \in \arg\min_{\hat{p}} \text{KL}(p||\hat{p})$, which we approximate with gradient descent. The matrices $\varphi$ and $\psi$ are initialized by drawing weights randomly from a normal distribution with a very small standard deviation.

We focus solely on value estimation in the Garnet MDPs with a fixed policy to simplify the experimental setup. We also use the ground truth reward function.

**Results:** The numerical results are graphed in Figure 2. For the near-deterministic ground truth environment, we see no benefit from using a calibrated $(m, b)$-VAML. We find that the algorithms are able to exploit the non-linear softmax function to achieve very accurate models in deterministic environments. However, even small amounts of stochasticity prevent this solution.

As the stochasticity increases, we see an advantage for calibrated losses. $(1, 1)$-VAML is not able to achieve good results even in a deterministic environment. As we initialize the models with high entropy, $(1, 1)$-VAML is unable to learn a correct value function. This corroborates the

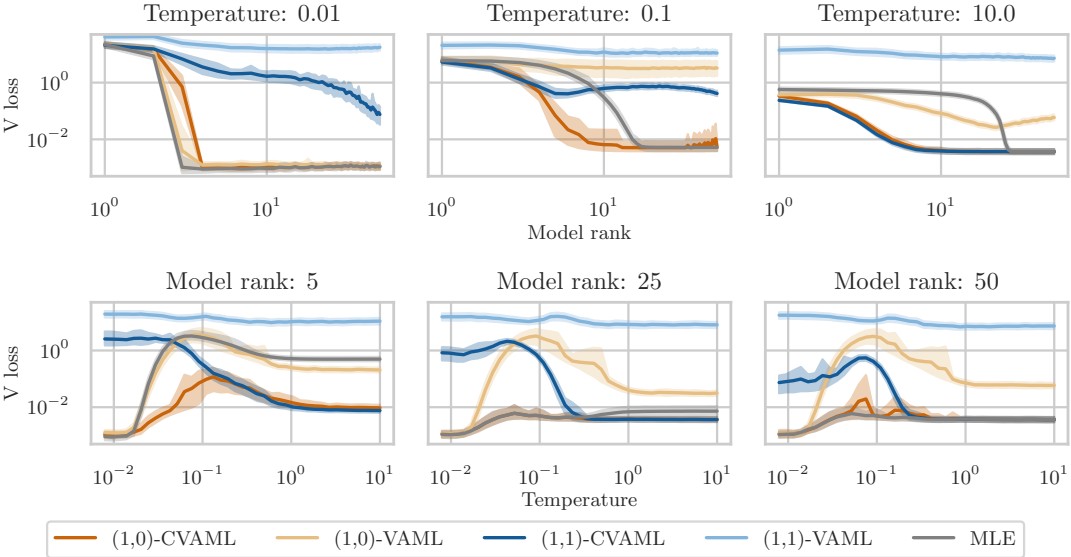

*Figure 2.* Results for the Garnet experiments. In the top row, we show the mean squared error of the value function prediction using different latent sizes $k$, over three different temperatures. In the bottom row, we vary the temperature and show results for three values of $k$. Shaded regions are bootstrapped confidence intervals of the mean at 95% over 1000 independent problems. With the exception of deterministic problems (left), the CVAML loss reliably results in the lowest value prediction error.

theoretical finding that the error depends on the model's stochasticity and not on the environment.

Calibrated $(1, 1)$-VAML still struggles in environments with low stochasticity. We find that the model is prone to get stuck in local plateaus as it reduces the value function difference per state faster than the model predictive variance. This suggests that a $(m \geq 1, 0)$-CVAML loss is preferable with stochastic environments.

## 6. Stochasticity and auxiliary models

All previous results hold independent of the model architecture. However, in practice, most implementations of decision-aware models use deterministic latent model structures (Schrittwieser et al., 2020; Ye et al., 2021; Hansen et al., 2022; Antonoglou et al., 2022).

In general, deterministic function approximations cannot capture the full transition distribution in stochastic environments. However, it is an open question whether a deterministic *latent* model can be sufficient for learning a value-equivalent model, as conjectured by Oh et al. (2017). We settle this question now.

### 6.1. Deterministic models for stochastic environments

Showing the existence of a deterministic value-aware model relies on the continuity of the transition kernel and involved functions $\varphi$ and $V$. These are standard assumptions that

are necessary to prove the existence and measurability of standard functions such as the value function (Bertsekas & Shreve, 1978).

**Proposition 5.** *Let $\mathcal{X}$ be a compact, connected, metrizable space. Let $p$ be a continuous kernel from $\mathcal{X}$ to probability measures over $\mathcal{X}$. Let $\mathcal{Z}$ be a metrizable space. Consider a bijective latent mapping $\phi : \mathcal{X} \to \mathcal{Z}$ and any $V : \mathcal{Z} \to \mathbb{R}$. Assume that they are both continuous. Denote $V_{\mathcal{X}} = V \circ \phi$.*

*Then there exists a measurable function $f^* : \mathcal{Z} \to \mathcal{Z}$ such that we have $V(f^*(\phi(x))) = \mathbb{E}_p \left[ V_{\mathcal{X}}(x^{(1)}) \right]$ for all $x \in \mathcal{X}$.*

*Furthermore, the same $f^*$ is a minimizer of the expected IterVAML loss over any distribution $x \sim \rho$*

$$f^* \in \arg\min_{\hat{f}} \mathbb{E}_{\rho, \mathcal{P}^\pi} \left[ \mathcal{L}_{\text{IterVAML},1}(\hat{f}, V_{\mathcal{X}} | V_{\mathcal{X}}, x) \right].$$

Proofs for this section can be found in Subsection A.3.

We can conclude that given a sufficiently flexible function class $\mathcal{F}$, $(1, b)$-VAML can recover an optimal deterministic model for value function prediction. Note that our conditions solely ensure the *existence* of a measurable function; the learning problem might still be very challenging.

Note that the conditions for our proof here are slightly different than those for our definition of an *uncalibrated* loss. Here we show that given a flexible enough function class, a deterministic model can sufficiently minimize the Iter-VAML loss. In the previous section, we showed that if the

function class admits a perfect stochastic model, but not a perfect deterministic one, the $(m, b)$-VAML loss will incur a calibration error. Here we show the conditions under which a perfect deterministic model exists.

Therefore, it is still important to have a calibrated surrogate loss for stochastic models, as many use cases make stochastic models attractive. For example, a pre-trained LLM backbone might be used, which is naturally stochastic due to the sampling strategies used to query it.

### 6.2. Auxiliary losses for latent-space models

While the original MuZero algorithm only uses the MuZero loss to update the latent embedding, dynamics model, and value function estimation, several more recent works add auxiliary stabilizing losses (Ye et al., 2021; Hafner et al., 2021; Hansen et al., 2024; Voelcker et al., 2025). These allow the model to learn meaningful transitions even before the value function is properly approximated, which helps e.g. in sparse-reward environments.

Most prior works use some form of Bootstrap-your-own-latent (BYOL) (Grill et al., 2020) loss, which minimizes the difference between the next states encoding $\varphi(x^{(m)})$ and the model prediction $f^m(x^{(0)})$. In the following we write $(\hat{f}, \varphi) = \hat{p}$ when we want to highlight the different components of the learned model.

In the case of linear features, models, and value functions, recent works have shown that such a loss can greatly aid in learning meaningful features for value function prediction (Lyle et al., 2021; Tang et al., 2023; Ni et al., 2024; Voelcker et al., 2024). When the difference is computed with an $L_2$, the auxiliary loss is

$$\hat{\mathcal{L}}_{\text{aux},m}^{\mathcal{P}^\pi}\left((\hat{f}, \varphi)|x\right) = \left\|\hat{f}^m(\varphi(x)) - \left[\varphi(x^{(m)})\right]_{\text{sg}}\right\|^2. \quad (6)$$

The introduction of a stabilizing loss poses a challenge for the result in Proposition 5. For a flexible enough model and embedding function class, the perfect model $f_{\text{aux}}^*$ would predict $\mathbb{E}_{\mathcal{P}^\pi}[\varphi(x')|x]$. However, $f_{\text{aux}}^*$ only coincides with the optimal model under the VAML loss if

$$\mathbb{E}_{\mathcal{P}^\pi}\left[\hat{V}(\varphi(x^{(1)}))\right] = \hat{V}\left(\mathbb{E}_{\mathcal{P}^\pi}\left[\varphi(x^{(1)})\right]\right). \quad (7)$$

This is the case if and only if the value function is affine in the embedding features $\varphi$, which is also referred to as a linear expectation model (Wan et al., 2019). However, learning an embedding in which the value function is linear can be difficult and may not lead to stable model predictions in complex, high-dimensional environments.

An exception to the use of deterministic models is the architecture proposed by Antonoglou et al. (2022). However,

their model and auxiliary loss rely on a biased straight-through gradient estimation. This introduces complications for finding the optima of the loss function and model class.

## 7. Experiments with latent-space models

We examine the impact of the calibrated losses on a subset of DMC environments (Tunyasuvunakool et al., 2020) encompassing 7 total tasks across the *humanoid* and *dog* domains. These two domains are the most challenging in the DMC suite, and the standard comparison for current methods (Voelcker et al., 2025; Nauman et al., 2024).

**Architecture and Setup:** We present a comparison between $(m, b)$-CVAML and $(m, b)$-VAML families of losses. To conduct our experiments we use two neural network architectures for the model, a stochastic and a deterministic latent model, while keeping the rest of the training setup consistent for a clean comparison. All experiments are conducted with latent model architectures composed of an encoder, a latent dynamics model, and a value and policy function head.

For the stochastic case, the latent models are multivariate, diagonal Gaussian distributions where mean and variance are parameterized by the latent network (Chua et al., 2018; Janner et al., 2019; Paster et al., 2021). In the deterministic case we simply use a feed-forward network.

Model rollouts are produced by sequentially sampling latent states from the model conditional on the initial state and an action sequence using the reparametrization trick.

$$\hat{p}(\hat{z}'|z, a) = \hat{\mu}(z, a) + \hat{\sigma}(z, a) \cdot \varepsilon, \ \varepsilon \sim \mathcal{N}(0, I) \quad (8)$$

The auxiliary loss for stochastic models is computed as a negative log-likelihood of the next states' latent representations under the current dynamics model.

$$\mathcal{L}_{\text{aux},m}\left(\hat{p}, \varphi\,\middle|\,x, a^{(0:m-1)}, x^{(m)}\right)$$
$$= -\log \hat{p}^m\left(\varphi(x^{(m)})\,\middle|\,\varphi(x), a^{(0:m-1)}\right), \quad (9)$$

where $a^{(0:m-1)}$ is a sequence of actions of length $m$ starting from state $x^{(0)}$. In the case of deterministic models, the auxiliary loss is simply the MSE version introduced in Equation 7.

The full model loss used is

$$\hat{\mathcal{L}}_{\text{model},m}^{\mathcal{P}^\pi}\left((\hat{f}, \varphi), \hat{V}|x\right)$$
$$= \underbrace{\hat{\mathcal{L}}_{m,b}^{\mathcal{P}^\pi}\left((\hat{f}, \varphi), \hat{V}|x\right)}_{(m,b)-\text{VAML}} + \underbrace{\hat{\mathcal{L}}_{\text{aux},m}^{\mathcal{P}^\pi}\left((\hat{f}, \varphi)|x\right)}_{\text{auxiliary}}, \quad (10)$$

replacing $\hat{f}$ with $\hat{p}$ for the stochastic model version.

For training the policy and value function, we follow the data-mixing protocol suggested by Voelcker et al. (2025).

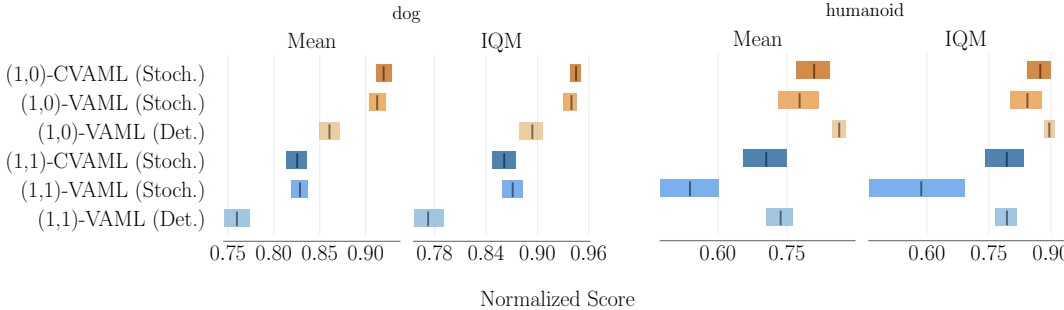

*Figure 3.* Results for the latent-space model experiments. We show the aggregate metrics of the final performance on dog (left) and humanoid (right) environments, following (Agarwal et al., 2021). Calibrating $(1, 1)$-VAML leads to a clear improvement in performance in the humanoid environment, while for $(1, 0)$-(C)VAML the difference is less noticeable. This is consistent with our theoretical findings as learning a smaller variance model can be sufficient, but wrong value functions will be learned with a stochastic model when $b \geq 1$.

We use both model-generated on-policy and real environment data from the replay buffer to train a value and policy head using the $(m, b)$-VAML loss with a twinned critic parametrization similar to TD3 (Fujimoto et al., 2018b). At every timestep, half of the next states in the minibatch are replaced with model-simulated states resulting from on-policy actions, the other half ground-truth environment data. This means the model influences the value function both through the encoder and through on-policy model-generated data.

The actions for environment interactions are obtained from the actor and the environment model using MPC (Hansen et al., 2022): initialized at the action produced by the actor for the current state, this algorithm iteratively refines the action to maximize the expected return. In total our setup uses the model for training the shared encoder, generating data for value and policy improvement, and for online model-based search.

For $(m, b)$-CVAML, the calibration term is computed by sampling multiple trajectories from the model and calculating the means and variances of next-state values produced by the critic across the different samples. Additional implementation details are provided in Appendix C.

**Results:** We plot aggregated performance over 20 random seeds with 95% CI, estimated with stratified percentile bootstrap (Patterson et al., 2024). Aggregations of final performance over several environments are visualized using the RLiable library (Agarwal et al., 2021).

The difference in performance is most noticeable between the $(1, 1)$-VAML and -CVAML losses on the humanoid benchmark, as the uncalibrated loss impacts both model and value function learning in this case. This effect is less prominent but a trend is still visible with $(1, 0)$-CVAML, where only the model learning is affected.

In the humanoid domain, we observe a performance improvement when using the calibration with the MuZero-

style loss. The dog domain is less affected by the difference in the calibration.

In addition to the calibration effect, we observe that probabilistic models outperform deterministic ones in the dog domain, even though the simulator is deterministic. This is in line with previous work (Chua et al., 2018; Janner et al., 2019), as stochastic models can reduce the tendency of the critic to exploit model errors. However, this advantage seems to be domain-specific and is less pronounced in the humanoid suite, where deterministic models with a $(1, 0)$-VAML loss perform best overall.

Finally, we observe an advantage of $(1, 0)$-updates over the $(1, 1)$-updates. While previous work claimed that IterVAML is unstable (Lovatto et al., 2020; Voelcker et al., 2022), comparisons were not made with SOTA model architectures and auxiliary tasks. As additional experiments (see Figure 8) show, the IterVAML loss is stable even without additional auxiliary losses, although adding the BYOL loss is necessary to achieve non-trivial performance in humanoid tasks.

> **Insight 2** (Remarks for practitioners). *While deterministic models are theoretically sufficient for learning value-equivalent models, we observe benefits in some benchmarks from the use of stochastic models. Using calibrated losses is empirically especially important for MuZero-style model and value updates.*

## 8. Related work

**VAML and MuZero:** Farahmand (2018) established Iter-VAML based on earlier work (Farahmand et al., 2017). Several extensions have been proposed, such as a VAML-regularized MSE loss (Voelcker et al., 2022) and a policy-gradient aware loss (Abachi et al., 2020). Combining Iter-VAML with latent spaces was first explored by Abachi et al. (2022), but no experimental results were provided.

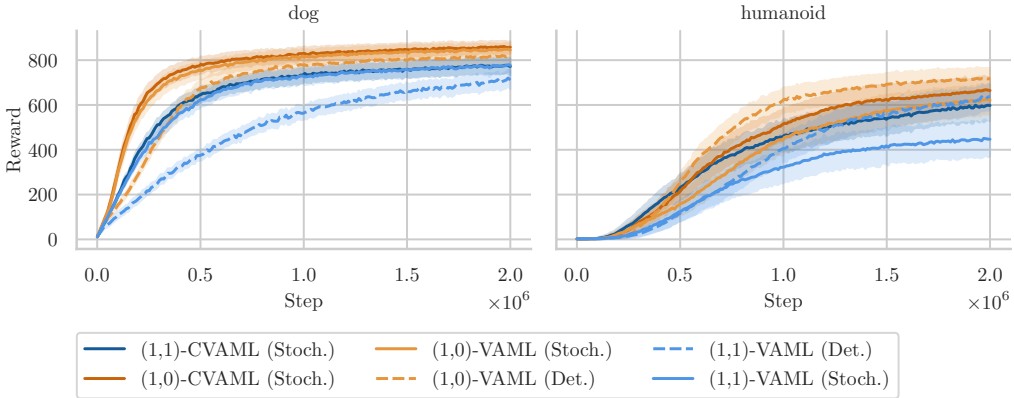

*Figure 4.* Sample efficiency curve for both dog (left) and humanoid (right). Per-task normalized return is aggregated over 30 seeds per environment and 3 tasks for the humanoid domain, and 4 – for the dog domain, with 95% bootstrapped confidence intervals shaded. In addition to a higher final return, we observe significantly earlier learning for the $(1, 0)$-(C)VAML on the dog task.

MuZero (Schrittwieser et al., 2020; Ye et al., 2021) is built based on earlier works that introduce the ideas of learning a latent model jointly with the value function (Silver et al., 2017; Oh et al., 2017). However, none of these works investigate the calibration of the loss function. Antonoglou et al. (2022) propose an extension to MuZero in stochastic environments but focus on the model architecture, not the value function loss. Hansen et al. (2022) and Hansen et al. (2024) adapted the MuZero loss to continuous control environments but did not extend their formulation to stochastic models. Grimm et al. (2020) and Grimm et al. (2021) consider how the set of value equivalent models relates to value functions. They are the first to show the close connection between the notions of value-awareness and MuZero.

**Other decision-aware algorithms:** Several other works propose decision-aware model learning algorithms that do not directly minimize a value function difference. D'Oro et al. (2020) weigh the samples used for model learning by their impact on the policy gradient. Nikishin et al. (2021) uses implicit differentiation to obtain a loss for the model function with regard to the policy performance measure. To achieve the same goal, Eysenbach et al. (2022) and Ghugare et al. (2023) choose a variational formulation. Modhe et al. (2021) proposes to compute the advantage function resulting from different models instead of using the value function. Ayoub et al. (2020) presents an algorithm for selecting models based on their ability to predict value function estimates and provide regret bounds with this algorithm.

**Learning with suboptimal models:** Several works have focused on the broader goal of using models with errors without addressing the loss functions of the model. Among these, some attempt to correct models using information obtained during exploration (Joseph et al., 2013; Talvitie, 2017; Modi et al., 2020; Rakhsha et al., 2022; 2024), or to

limit interaction with wrong models (Buckman et al., 2018; Janner et al., 2019; Abbas et al., 2020). Several of these techniques can be applied together with $(m, b)$-CVAML to improve the model and value function learning further. Finally, we do not focus on exploration, but Guo et al. (2022) show how auxiliary losses can be used not only to stabilize learning but also to improve exploration.

## 9. Conclusions

We theoretically analyze commonly used value-aware losses such as the MuZero and IterVAML loss and show that they are *uncalibrated* surrogate losses. When using $(m, b)$-VAML losses, such as the popular IterVAML and MuZero algorithms, with stochastic environment models, the loss learns low variance models, even if those do not recover the correct value function. Building on our proofs, we propose a novel variant of the loss to stabilize learning with stochastic environments and evaluate its efficacy in practice.

Our experiments further show that the calibration of the $(m, b)$-VAML losses is important for obtaining strong learning with stochastic environment models. In addition, while previous papers showed that IterVAML losses can be unstable in practice (Lovatto et al., 2020; Voelcker et al., 2022), we find that this can be overcome by adopting the latent model architecture used by Schrittwieser et al. (2020) and the auxiliary losses established by Li et al. (2023); Hansen et al. (2022). When combined with a suitable value learning procedure, IterVAML performs on par with or better than MuZero in continuous control tasks. Finally, while $(m, b)$-VAML losses have mostly been used with deterministic environments, our work enables the community to use stochastic models with a calibrated loss and shows the potential merits of this approach in a number of environments.

## Acknowledgments

We would like to thank the members of the Adaptive Agents Lab and the Toronto Intelligent Systems Lab who provided feedback on the ideas, the draft of this paper, and provided additional resources for running all experiments. We acknowledge Amin Rakhsha for helping with the formal statement of Lemma 4. We would also like to thank the anonymous reviewers for providing valuable feedback to improve this work. AMF acknowledges the funding from the Natural Sciences and Engineering Research Council of Canada (NSERC) through the Discovery Grant program (2021-03701). Resources used in preparing this research were provided, in part, by the Province of Ontario, the Government of Canada through CIFAR, and companies sponsoring the Vector Institute.

## Impact Statement

Our paper is first and foremost a theoretical analysis of loss function used in reinforcement learning. As such, it presents work whose goal is to advance the field of Machine Learning. There are many potential societal consequences of our work, none of which we feel must be specifically highlighted here.

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

# A. Proofs and mathematical clarifications

We provide helper lemmata for Section 4 and Section 6 in this section.

All of our proofs rely heavily on a standard expansion technique which is used to prove that sample-based losses correctly approximate population losses. This is found in standard textbooks on learning theory such as Györfi et al. (2002).

It proceeds by expanding a loss $\mathbb{E}_{x \sim p, y \sim q(\cdot|x)} \left[ |f(x) - y|^2 \right]$ with an expected target $f^*(x)$. In our case, this is mostly the minimizer of the target loss when evaluating surrogate losses. Then we obtain

$$\mathbb{E}_{x \sim p, y \sim q(\cdot|x)} \left[ |f(x) - y|^2 \right] = \mathbb{E}_{x \sim p, y \sim q(\cdot|x)} \left[ |f(x) - f^*(x) + f^*(x) - y|^2 \right],$$

and continue expanding from there. Issues generally arise when $y$ depends on $f(x)$ in some way, or when $f(x)$ itself involves a sampling procedure.

## A.1. Main propositions: Section 3

**Proposition 1.** *Farahmand (2018) Let $\mathcal{P}^\pi$ be the policy-conditioned transition kernel of an MDP and let $V : \mathcal{X} \to \mathbb{R}$ be a function. Let $\hat{p}(\cdot|x)$ be a model so that $\mathcal{L}_{\mathrm{IterVAML},1}^{\mathcal{P}^\pi}(\hat{p}, V|x) = 0$. Then $[\mathcal{T}_{\mathcal{P}^\pi} V](x) = [\mathcal{T}_{\hat{p}} V](x)$.*

*Proof.* Assume $\hat{p}$ fulfills $\mathcal{L}_{\mathrm{IterVAML},1}^{\mathcal{P}^\pi}(\hat{p}, \hat{V}|x) = 0$. Then $\mathbb{E}_{\hat{p}}[V(\hat{x}^{(1)})] = \mathbb{E}_{\mathcal{P}^\pi}[V(x^{(1)})]$. By definition of the Bellman operator, we have

$$[\mathcal{T}^{\mathcal{P}^\pi} V](x) = r(x) + \gamma \mathbb{E}_{x^{(1)} \sim \mathcal{P}^\pi(\cdot|x)}[V(x^{(1)})] = r(x) + \gamma \mathbb{E}_{x^{(1)} \sim \hat{p}(\cdot|x)}[V(x^{(1)})] = [\mathcal{T}^{\hat{p}} V](x). \tag{11}$$

$\square$

## A.2. Main propositions: Section 4

### A.2.1. HELPER LEMMATA

For the following results, we will use the following notation. Let $\mathcal{X}$ be a discrete sample space and $p$ a distribution over it. Let $f : \mathcal{X} \to \mathbb{R}$ be a random variable. Let $\mathcal{P}^*$ be the set of distributions for which both $\mathcal{P}^* = \mathrm{Arg}\min_{p'} \mathrm{Var}_{p'}[f(x)]$, and for all $p^* \in \mathcal{P}^*$ $\mathbb{E}_{p^*}[f(x)] = \mathbb{E}_p[f(x)]$. Let $k$ be an integer. Finally, let $g(\xi) = \mathbb{E}_{x \sim p}\left[(\mathbb{E}_{y \sim \xi}[f(y)] - f(x))^2\right] + \frac{1}{k}\mathrm{Var}_\xi[f(x)]$.

We assume the following condition on a distribution $p$ and a function $f$ for all the lemmata in this section.

**Assumption 1.** *Let $p$ be a probability distribution over $\mathcal{X}$ with $\mathbb{E}_p[f(x)]$, for which no $x$ exists so that $f(x) = \mathbb{E}_p[f(x)]$. This is an assumption on both $f$ and $p$.*

This is an important assumption as it guarantees that there is no distribution $q$ with $0$ variance such that $\mathbb{E}_q[f(x)] = \mathbb{E}_p[f(x)]$. This excludes a corner case of our proof, as fully deterministic environments and models do not lead to an uncalibrated IterVAML loss.

We now obtain three simple lemmata about the minima of the function $g$.

**Lemma 1.** *There does not exist a distribution $p'$ such that $p' \notin \mathcal{P}^*$, $\mathbb{E}_{p'}[f(x)] = \mathbb{E}_p[f(x)]$, and $g(p') \leq g(p^*)$ for any $p^* \in \mathcal{P}$.*

*Proof.* To prove the lemma, we first evaluate $g$ for any distribution $\xi^*$ with $\mathbb{E}_{\xi^*}[f(x)] = \mathbb{E}_p[f(x)]$

$$g(\xi^*) = \mathbb{E}_{x \sim p}\left[(\mathbb{E}_{y \sim \xi^*}[f(y)] - f(x))^2\right] + \frac{1}{k}\mathrm{Var}_{\xi^*}[f(x)] \tag{12}$$

$$= \mathbb{E}_{x \sim p}\left[(\mathbb{E}_{y \sim p}[f(y)] - f(x))^2\right] + \frac{1}{k}\mathrm{Var}_{\xi^*}[f(x)] \tag{13}$$

$$= \mathrm{Var}_p(f(x)) + \frac{1}{k}\mathrm{Var}_{\xi^*}[f(x)]. \tag{14}$$

As we constructed the set $\mathcal{P}^*$ so that all members have equal (minimum) variance, we obtain the same $g(p^*)$ for all $p^* \in \mathcal{P}^*$.

We now set up a contradiction by assuming $p'$ exists. By this assumption

$$g(p') < g(p^*) \tag{15}$$

$$\text{Var}_p(f(x)) + \frac{1}{k}\text{Var}_{p'}[f(x)] < \text{Var}_p(f(x)) + \frac{1}{k}\text{Var}_{p^*}[f(x)] \tag{16}$$

$$\text{Var}_{p'}[f(x)] < \text{Var}_{p^*}[f(x)] \tag{17}$$

For the function $p'$ to exist so that $g(p') < g(p^*)$, we would therefore require $\text{Var}_{p'}[f(x)] < \text{Var}_{p^*}[f(x)]$, which is impossible by the definition of $\mathcal{P}^*$. This is a contradiction with the requirements for $p'$ in the lemma, which concludes our proof. $\square$

In the following lemma, we show that under some conditions a distribution $q$ exists which has $g(q) < g(p^*)$ and for which $\mathbb{E}_p[f(x)] \neq \mathbb{E}_q[f(x)]$. We can rewrite $g$ as

$$g(q) = \mathbb{E}_p\left[(\mathbb{E}_q[f(y)] - f(x))^2\right] + \frac{1}{k}\text{Var}_q[f(x)] \tag{18}$$

$$= \mathbb{E}_p[f(x)^2] - 2\mathbb{E}_p[f(x)]\mathbb{E}_q[f(x)] + \mathbb{E}_q[f(x)]^2 + \frac{1}{k}\text{Var}_q[f(x)] \tag{19}$$

$$\leq \mathbb{E}_p[f(x)^2] - 2\mathbb{E}_p[f(x)]\mathbb{E}_q[f(x)] + \mathbb{E}_q[f(x)^2] + \frac{1}{k}\text{Var}_q[f(x)] \tag{20}$$

$$= \text{Var}_p[f(x)] + \frac{k+1}{k}\text{Var}_q[f(x)] + (\mathbb{E}_p[f(x)] - \mathbb{E}_q[f(x)])^2 \tag{21}$$

Equation 20 follows from Jensen's inequality. Intuitively, the function $g$ depends on the squared deviation of the expectation and the variance of both $p$ and $q$. If the variance of $q$ can be reduced more than the squared deviation of the means, then $\mathcal{P}^*$ will not contain the minimum of $g$.

Note that the conditions on $q$ are sufficient but not necessary, as we use Jensen's inequality to obtain our bound. In addition, to simplify the proof, we assume that $q$ is a distribution with zero variance. In practice, any distribution with $\frac{k+1}{k}\text{Var}_q[f(x)] + (\mathbb{E}_p[f(x)] - \mathbb{E}_q[f(x)])^2 < \text{Var}_{p^*}[f(x)]$ and $\mathbb{E}_q[f(x)] \neq \mathbb{E}_p[f(x)]$ will suffice, but this requirement is somewhat less intuitive to grasp. As our definition of calibration does not require us to exhaustively characterize all cases for $p$, $p^*$, and $q$, we have chosen this set of conditions which simplifies the proof. As our Garnet experiments show, many randomly generated transition distributions admit distributions where the minimizer of $g$ does not match $p$ in expectation.

**Lemma 2.** *If there exists a distribution $q$ with $(\mathbb{E}_q[f(x)] - \mathbb{E}_p[f(x)])^2 < \frac{1}{k}\text{Var}_{p^*}[f(x)]$, and $\text{Var}_q[f(x)] = 0$, then $g(q) < g(p^*)$ for all $p^*$ and, by the assumptions on $p$ and $f$, $\mathbb{E}_{x\sim q}[f(x)] \neq \mathbb{E}_{x\sim p}[f(x)]$.*

*Proof.* Choose any $p^* \in \mathcal{P}^*$. As $\text{Var}_q[f(x)] = 0$, $g(q) = \mathbb{E}_{x\sim p}[(\mathbb{E}_{y\sim q}[f(y)] - f(x))]$. We can now decompose $g(q)$ as

$$g(q) = \mathbb{E}_p\left[(\mathbb{E}_q[f(y)] - f(x))^2\right] = \mathbb{E}_p[f(x)^2] - 2\mathbb{E}_p[f(x)]\mathbb{E}_q[f(x)] + \mathbb{E}_q[f(x)]^2 \tag{22}$$

$$= \text{Var}_p[f(x)] + (\mathbb{E}_p[f(x)] - \mathbb{E}_q[f(x)])^2 \tag{23}$$

$$< \text{Var}_p[f(x)] + \frac{1}{k}\text{Var}_{p^*}[f(x)] = g(p^*). \tag{24}$$

Equation 24 follows from the assumption on q.

By the assumptions on $p$ and $f$ there does not exist a $x \in \mathcal{X}$ so that $f(x) = \mathbb{E}_p[f(x)]$. However, as $\text{Var}_q[f(x)] = 0$, $\mathbb{E}_q[f(x)] = f(x_i)$ for some $x_i \in \mathcal{X}$. Therefore, $\mathbb{E}_q[f(x)] \neq \mathbb{E}_p[f(x)]$, which concludes the proof. $\square$

As a consequence, we obtain the following, final lemma.

**Lemma 3.** *Let $\mathcal{P}^{\mathbb{E}}$ be the set of all distributions $\xi$ so that $\mathbb{E}_p[f(x)] = \mathbb{E}_\xi[f(x)]$. Assume $q$ satisfying Lemma 2 exists. Then $\arg\min_\xi g(\xi) \not\subseteq \mathcal{P}^{\mathbb{E}}$.*

*Proof.* The lemma is a direct consequence of Lemma 1 and Lemma 2. By Lemma 1, for all $p' \in \mathcal{P}^{\mathbb{E}}$ $g(p') \geq g(p^*)$. As a consequence of Lemma 2, there exists at least one $q$ with $g(q) < g(p^*)$ and that $q$ has $\mathbb{E}_q[f(x)] \neq \mathbb{E}_p[f(x)]$. Then the statement follows directly as $q \notin \mathcal{P}^{\mathbb{E}}$. $\square$

A.2.2. MAIN RESULTS: ITERVAML

**Proposition 2.** *Let* $\hat{\mathcal{L}}^{\mathcal{P}^\pi}_{m\geq 1,0}(\hat{p}, V|x, x^{(m)})$ *be the surrogate loss for* $\mathcal{L}^{\mathcal{P}^\pi}_{\text{IterVAML},m}(\hat{p}, V|x)$. *Let* $\mathcal{P}^*$ *be the set of all distributions* $p$ *for which* $\mathbb{E}_p\left[V(\hat{x}^{(m)})\right] = \mathbb{E}_{\mathcal{P}^\pi}\left[V(x^{(m)})\right]$. *There exist an MDP and a class of distributions* $\mathbb{P}$ *with* $\mathcal{P}^\pi \in \mathcal{P}^* \subseteq \mathbb{P}$ *so that*

$$\underset{\hat{p} \in \mathbb{P}}{\text{Arg min}}\, \mathbb{E}_{\hat{p}}\left[\hat{\mathcal{L}}^{\mathcal{P}^\pi}_{m\geq 1,0}(\hat{p}, V|x, x^{(m)})\right] \not\subseteq \mathcal{P}^*.$$

*Therefore,* $\hat{\mathcal{L}}_{\text{model}}$ *is minimum-uncalibrated.*

*Proof.* Expanding the empirical IterVAML loss with $k$ samples, obtain

$$\mathbb{E}_{\mathcal{P}^\pi}\left[\hat{\mathcal{L}}^{\mathcal{P}^\pi}_{m\geq 1,0}\left(\hat{p}, V|x, x^{(m)}\right)\right] \tag{25}$$

$$= \mathbb{E}_{\hat{p}, \mathcal{P}^\pi}\left[\left(V\left(\hat{x}^{(m)}\right) - V\left(x^{(m)}\right)\right)^2\right] \tag{26}$$

$$= \mathbb{E}_{\hat{p}, \mathcal{P}^\pi}\left[\left(V\left(\hat{x}^{(m)}\right) - \mathbb{E}_{\hat{p}}\left[V\left(\hat{x}^{(m)}\right)\right] + \mathbb{E}_{\hat{p}}\left[V\left(\hat{x}^{(m)}\right)\right] - V\left(x^{(m)}\right)\right)^2\right] \tag{27}$$

$$= \mathbb{E}_{\hat{p}, \mathcal{P}^\pi}\left[\left(V\left(\hat{x}^{(m)}\right) - \mathbb{E}_{\hat{p}}\left[V\left(\hat{x}^{(m)}\right)\right]\right)^2\right] + \tag{28}$$

$$\underbrace{2\,\mathbb{E}_{\hat{p}, \mathcal{P}^\pi}\left[\left(V\left(\hat{x}^{(m)}\right) - \mathbb{E}_{\hat{p}}\left[V\left(\hat{x}^{(m)}\right)\right]\right)\left(\mathbb{E}_{\hat{p}, \mathcal{P}^\pi}\left[V\left(\hat{x}^{(m)}\right)\right] - V\left(x^{(m)}\right)\right)\right]}_{=0} + \tag{29}$$

$$\mathbb{E}_{\hat{p}, \mathcal{P}^\pi}\left[\left(\mathbb{E}_{\hat{p}}\left[V\left(\hat{x}^{(m)}\right)\right] - V\left(x^{(m)}\right)\right)^2\right] \tag{30}$$

$$= \underbrace{\mathbb{E}_{\hat{p}, \mathcal{P}^\pi}\left[\left(V\left(\hat{x}^{(m)}\right) - \mathbb{E}_{\hat{p}}\left[V\left(\hat{x}^{(m)}\right)\right]\right)^2\right]}_{=\text{Var}} + \underbrace{\mathbb{E}_{\hat{p}, \mathcal{P}^\pi}\left[\left(\mathbb{E}_{\hat{p}}\left[V\left(\hat{x}^{(m)}\right)\right] - V\left(x^{(m)}\right)\right)^2\right]}_{(2)} \tag{31}$$

Equation 29 is 0 since $\mathbb{E}_{\hat{p}, \mathcal{P}^\pi}\left[V\left(\hat{x}^{(m)}\right) - \mathbb{E}_{\hat{p}}\left[V\left(\hat{x}^{(m)}\right)\right]\right] = 0$, and samples from $\hat{p}$ and $\mathcal{P}^\pi$ are independent. Note that we can separate the first factor and the second factor since the first one only contains $\hat{X}^{(m)}$ as a random variable, and the second one only contains $x^{(m)}$. The second factor only includes $\hat{x}^{(m)}$ inside of an expectation, and the expected value is not a random variable anymore.

The final term can again be decomposed as

$$\mathbb{E}_{\hat{p}, \mathcal{P}^\pi}\left[\left(\mathbb{E}_{\hat{p}}\left[V\left(\hat{x}^{(m)}\right)\right] - V\left(x^{(m)}\right)\right)^2\right] \tag{32}$$

$$= \mathbb{E}_{\hat{p}, \mathcal{P}^\pi}\left[\left(\mathbb{E}_{\hat{p}}\left[V\left(\hat{x}^{(m)}\right)\right] - \mathbb{E}_{\mathcal{P}^\pi}\left[V\left(x^{(m)}\right)\right] + \mathbb{E}_{\mathcal{P}^\pi}\left[V\left(x^{(m)}\right)\right] - V\left(x^{(m)}\right)\right)^2\right] \tag{33}$$

$$= \underbrace{\mathbb{E}_{\hat{p}, \mathcal{P}^\pi}\left[\left(\mathbb{E}_{\hat{p}}\left[V\left(\hat{x}^{(m)}\right)\right] - \mathbb{E}_{\mathcal{P}^\pi}\left[V\left(x^{(m)}\right)\right]\right)^2\right]}_{=\text{IterVAML}} + \underbrace{\mathbb{E}_{\hat{p}, \mathcal{P}^\pi}\left[\left(\mathbb{E}_{\mathcal{P}^\pi}\left[V\left(x^{(m)}\right)\right] - V\left(x^{(m)}\right)\right)^2\right]}_{\text{independent of } \hat{p}}. \tag{34}$$

The middle term of the binomial expansion again vanishes as $\mathbb{E}_{\mathcal{P}^\pi}\left[\mathbb{E}_{\mathcal{P}^\pi}\left[V\left(x^{(m)}\right)\right] - V\left(x^{(m)}\right)\right] = 0$, and samples from $\hat{p}$ and $\mathcal{P}^\pi$ are independent.

When we drop the term that is independent of the model, the loss has the form of $g$ in Lemma 3

$$\mathbb{E}_{\mathcal{P}^\pi}\left[\hat{\mathcal{L}}^{\mathcal{P}^\pi}_{m\geq 1,0}\left(\hat{p}, V|x, x^{(m)}\right)\right] \tag{35}$$

$$= \underbrace{\mathbb{E}_{\hat{p}, \mathcal{P}^\pi}\left[\left(V\left(\hat{x}^{(m)}\right) - \mathbb{E}_{\hat{p}}\left[V\left(\hat{x}^{(m)}\right)\right]\right)^2\right]}_{=\text{Var}} + \underbrace{\mathbb{E}_{\hat{p}, \mathcal{P}^\pi}\left[\left(\mathbb{E}_{\hat{p}}\left[V\left(\hat{x}^{(m)}\right)\right] - \mathbb{E}_{\mathcal{P}^\pi}\left[V\left(x^{(m)}\right)\right]\right)^2\right]}_{=\text{IterVAML}}. \tag{36}$$

Let $\mathcal{P}^\pi(\cdot|x)$ be a distribution so that Lemma 3 holds. Then there exists a $\hat{p}$ with $\hat{\mathcal{L}}_{m\geq 1,0}^{\mathcal{P}^\pi}\left(\hat{p}, V|x, x^{(m)}\right) < \hat{\mathcal{L}}_{m\geq 1,0}^{\mathcal{P}^\pi}\left(\mathcal{P}^\pi, V|x, x^{(m)}\right)$ and $\mathbb{E}_{\hat{p}}\left[V\left(\hat{x}^{(m)}\right)\right] \neq \mathbb{E}_{\mathcal{P}^\pi}\left[V(x^{(m)})\right]$. $\qquad\square$

If we use the $k$ sample version of the empirical IterVAML loss instead, we obtain the following decomposition

$$\mathbb{E}_{\mathcal{P}^\pi}\left[\hat{\mathcal{L}}_{m\geq 1,0}^{\mathcal{P}^\pi}\left(\hat{p}, V|x, x^{(m)}\right)\right] \tag{37}$$

$$= \underbrace{\frac{1}{k}\mathbb{E}_{\hat{p},\mathcal{P}^\pi}\left[\left(V\left(\hat{x}^{(m)}\right) - \mathbb{E}_{\hat{p}}\left[V\left(\hat{x}^{(m)}\right)\right]\right)^2\right]}_{=\frac{1}{k}\mathrm{Var}} + \underbrace{\mathbb{E}_{\hat{p},\mathcal{P}^\pi}\left[\left(\mathbb{E}_{\hat{p}}\left[V\left(\hat{x}^{(m)}\right)\right] - V\left(x^{(m)}\right)\right)^2\right]}_{\mathrm{IterVAML}}. \tag{38}$$

The only difference when using more samples is that the variance term is scaled by the factor $\frac{1}{k}$, as the variance of the mean estimator $\sum_{i=1}^k V\left(x_i^{(m)}\right)$ decreases. Consequently, for larger values of $k$, the condition in Lemma 2 for $g$ becomes stricter. As $k \to \infty$, the condition becomes unfulfillable. This is also intuitive, as $\frac{1}{k}\sum_{i=1}^k V\left(x_i^{(m)}\right) \to \mathbb{E}_{\hat{p}}\left[V\left(x^{(m)}\right)\right]$ as $k \to \infty$ almost surely (assuming standard conditions hold).

**Proposition 3.** *The variance-corrected loss* $\hat{\mathcal{L}}_{\mathrm{CVAML},m}^k(\hat{P}, V|x, x^{(m)})$ *is a calibrated surrogate loss for* $\mathcal{L}_{\mathrm{IterVAML},m}^{\mathcal{P}^\pi}(\hat{P}, V|x, x^{(m)})$.

*Proof.* Reusing the previous derivation, we obtain

$$\mathbb{E}_{\mathcal{P}^\pi}\left[\hat{\mathcal{L}}_{\mathrm{var},m}^k(\hat{P}, V|x, x^{(m)})\right] = \mathbb{E}_{\mathcal{P}^\pi}\left[\hat{\mathcal{L}}_{i\geq 1,0}^{\mathcal{P}^\pi}\left(\hat{p}, V|x, x^{(m)}\right)\right] - \frac{1}{k}\underbrace{\mathbb{E}_{\hat{x},\mathcal{P}^\pi}\left[\left(V\left(\hat{x}^{(m)}\right) - \mathbb{E}_{\hat{p}}\left[V\left(\hat{x}^{(m)}\right)\right]\right)^2\right]}_{=\mathrm{Var}} \tag{39}$$

$$= \underbrace{\mathbb{E}_{\hat{p},\mathcal{P}^\pi}\left[\left(\mathbb{E}_{\hat{p}}\left[V\left(\hat{x}^{(m)}\right)\right] - \mathbb{E}_{\mathcal{P}^\pi}\left[V\left(x^{(m)}\right)\right]\right)^2\right]}_{\mathrm{IterVAML}} \tag{40}$$

Therefore, the surrogate loss and target loss are equivalent in expectation. $\qquad\square$

### A.2.3. MAIN RESULTS: MUZERO

The following lemma shows that a function that minimizes a quadratic and a variance term cannot be the minimum function of the quadratic. This is used to show that the minimum of the MuZero value function learning term is not the same as applying the model-based Bellman operator.

**Lemma 4.** *Let* $g : \mathcal{X} \to \mathbb{R}$ *be a function that is not constant and let* $\mu$ *be a non-degenerate probability distribution over* $\mathcal{X}$. *Let* $\mathcal{L}(f) = \mathbb{E}_{x\sim\mu}\left[(f(x) - g(x))^2\right] + \mathbb{E}_{x\sim\mu}[f(x)g(x)] - \mathbb{E}_{x\sim\mu}[f(x)]\mathbb{E}_\mu[g(x)]$. *There exists a function space* $\mathcal{F}$ *with* $g \in \mathcal{F}$ *so that* $g \notin \arg\min_{f\in\mathcal{F}}\mathcal{L}(f)$.

*Proof.* The proof follows by showing that there is a descent direction from $g$ that improves upon $\mathcal{L}$. For this, we construct the auxiliary function $\hat{g}(x) = g(x) - \varepsilon g(x)$. Evaluating $\mathcal{L}(\hat{g})$ yields

$$\begin{aligned} &\varepsilon^2\mathbb{E}_\mu\left[g(x)^2\right] + \mathbb{E}_\mu\left[(g(x) - \varepsilon g(x))g(x)\right] \\ &- \mathbb{E}_\mu\left[(g(x) - \varepsilon g(x))\right]\mathbb{E}_\mu[g(x)] \\ &= \varepsilon^2\mathbb{E}_\mu\left[g(x)^2\right] + (1-\varepsilon)\mathbb{E}_\mu\left[g(x)^2\right] - (1-\varepsilon)\mathbb{E}_\mu[g(x)]^2. \end{aligned}$$

Taking the derivative of this function wrt to $\varepsilon$ yields

$$\begin{aligned} &\frac{\mathrm{d}}{\mathrm{d}\varepsilon}\varepsilon^2\mathbb{E}_\mu\left[g(x)^2\right] + (1-\varepsilon)\mathbb{E}_\mu\left[g(x)^2\right] - (1-\varepsilon)\mathbb{E}_\mu[g(x)]^2 \\ &= 2\varepsilon\,\mathbb{E}_\mu\left[g(x)^2\right] - \mathbb{E}_\mu\left[g(x)^2\right] + \mathbb{E}_\mu[g(x)]^2. \end{aligned}$$

Setting $\varepsilon$ to 0, we obtain

$$\mathbb{E}_\mu\left[g(x)\right]^2 - \mathbb{E}_\mu\left[g(x)^2\right] = \text{Var}_\mu\left[g(x)\right]$$

By the Cauchy-Schwarz inequality, the variance is only 0 for a $g(x)$ constant almost everywhere. However, this violates the assumption. Therefore there exists an $\varepsilon > 0$ so that $\mathcal{L}(\hat{g}) \leq \mathcal{L}(g)$. We now can construct the function space $\mathcal{F}$ so that it includes at least $g$ and $g + \varepsilon g$.

$\square$

**Proposition 4.** *Let $\mathcal{L}_{\text{TD}}(V|V_{\text{tar}}, x^{(m)})$ be the target loss, and let $\hat{\mathcal{L}}_{m,1}^{\mathcal{P}^\pi}(\mathcal{P}^\pi, V|V_{\text{tar}}, x, x^{(m),r^{(m)}})$ be the surrogate loss. There exists a set of functions $\mathcal{V}$ for any $V_{\text{tar}}$ that is not a constant function, for which two conditions hold:*

1. *The set is complete, meaning that $[\mathcal{T}_{\mathcal{P}^\pi} V_{\text{tar}}] \in \mathcal{V}$ for some target function $V_{\text{tar}}$,*

2. *and,*

$$\underset{\hat{V} \in \mathcal{V}}{\text{Arg min}}\, \mathbb{E}_{\mathcal{P}^\pi}\left[\hat{\mathcal{L}}_{m,1}^{\mathcal{P}^\pi}(\mathcal{P}^\pi, \hat{V}|V_{\text{tar}}, x, x^{(m)})\right] \nsubseteq [\mathcal{T}_{\mathcal{P}^\pi} V_{\text{tar}}].$$

*Therefore, $\hat{\mathcal{L}}_{m,1}^{\mathcal{P}^\pi}$ is minimum-uncalibrated.*

*Proof.* By assumption, let $\hat{p}$ in the MuZero loss be the true transition kernel $p$. Expand the MuZero loss by $\left[\mathcal{T}_{\mathcal{P}^\pi} V_{\text{tar}}\right]\left(\hat{x}^{(m)}\right)$ and take its expectation:

$$\mathbb{E}_{\mathcal{P}^\pi}\left[\hat{\mathcal{L}}_{m,b}^{\mathcal{P}^\pi}(\mathcal{P}^\pi, V|V_{\text{tar}}, x, x^{(m)})\right] \tag{41}$$

$$= \mathbb{E}_{\hat{p},\mathcal{P}^\pi}\left[\left[\hat{V}\left(\hat{x}^{(m)}\right) - \left[r\left(x^{(m)}\right) + \gamma V_{\text{tar}}\left(x^{(m+1)}\right)\right]\right]^2\right] \tag{42}$$

$$= \mathbb{E}_{\hat{p},\mathcal{P}^\pi}\left[\left[\hat{V}\left(\hat{x}^{(m)}\right) - \left[\mathcal{T}_{\mathcal{P}^\pi} V_{\text{tar}}\right]\left(\hat{x}^{(m)}\right) + \left[\mathcal{T}_{\mathcal{P}^\pi} V_{\text{tar}}\right]\left(\hat{x}^{(m)}\right) - \left[r\left(x^{(m)}\right) + \gamma V_{\text{tar}}\left(x^{(m+1)}\right)\right]\right]^2\right] \tag{43}$$

$$= \mathbb{E}_{\hat{p},\mathcal{P}^\pi}\left[\left(\hat{V}\left(\hat{x}^{(m)}\right) - \left[\mathcal{T}_{\mathcal{P}^\pi} V_{\text{tar}}\right]\left(\hat{x}^{(m)}\right)\right)^2\right] + \tag{44}$$

$$2\,\mathbb{E}_{\hat{p},\mathcal{P}^\pi}\left[\left(\hat{V}\left(\hat{x}^{(m)}\right) - \left[\mathcal{T}_{\mathcal{P}^\pi} V_{\text{tar}}\right]\left(\hat{x}^{(m)}\right)\right)\left(\left[\mathcal{T}_{\mathcal{P}^\pi} V_{\text{tar}}\right]\left(\hat{x}^{(m)}\right) - \left[r\left(x^{(m)}\right) + \gamma V_{\text{tar}}\left(x^{(m+1)}\right)\right]\right)\right] + \tag{45}$$

$$\mathbb{E}_{\hat{p},\mathcal{P}^\pi}\left[\left(\left[\mathcal{T}_{\mathcal{P}^\pi} V_{\text{tar}}\right]\left(\hat{x}^{(m)}\right) - \left[r\left(x^{(m)}\right) + \gamma V_{\text{tar}}\left(x^{(m+1)}\right)\right]\right)^2\right] \tag{46}$$

We aim to study the minimizer of this term. The first term (Equation 44) is the bootstrapped Bellman residual with a target $V_{\text{tar}}$. The third term (Equation 46) is independent of $\hat{V}$, so we can drop it when analyzing the minimization problem.

The second term (Equation 45) simplifies to

$$\mathbb{E}_{\hat{p},\mathcal{P}^\pi}\left[\hat{V}\left(\hat{x}^{(m)}\right)\left(\left[\mathcal{T}_{\mathcal{P}^\pi} V_{\text{tar}}\right]\left(\hat{x}^{(m)}\right) - \left[r\left(x^{(m)}\right) + \gamma V_{\text{tar}}\left(x^{(m+1)}\right)\right]\right)\right] \tag{47}$$

as the remainder is independent of $\hat{V}$ again.

This remaining term however is not independent of $\hat{V}$ and not equal to 0 either. Instead, it decomposes into a variance-like term, using the conditional independence of $\hat{x}^{(1)}$ and $x^{(1)}$ given $x^{(0)}$:

$$\mathbb{E}_{\hat{p},\mathcal{P}^\pi}\left[\hat{V}\left(\hat{x}^{(m)}\right)\left(\left[\mathcal{T}_{\mathcal{P}^\pi} V_{\text{tar}}\right]\left(\hat{x}^{(m)}\right) - \left[r\left(x^{(m)}\right) + \gamma V_{\text{tar}}\left(x^{(m+1)}\right)\right]\right)\right] \tag{48}$$

$$= \mathbb{E}_{\hat{p}}\left[\hat{V}\left(\hat{x}^{(m)}\right)\left[\mathcal{T}_{\mathcal{P}^\pi} V_{\text{tar}}\right]\left(\hat{x}^{(m)}\right)\right] - \mathbb{E}_{\hat{p},\mathcal{P}^\pi}\left[\hat{V}\left(\hat{x}^{(1)}\right)\left[r\left(x^{(m)}\right) + \gamma V_{\text{tar}}\left(x^{(m+1)}\right)\right]\right] \tag{49}$$

$$= \mathbb{E}_{\hat{p}}\left[\hat{V}\left(\hat{x}^{(m)}\right)\left[\mathcal{T}_{\mathcal{P}^\pi} V_{\text{tar}}\right]\left(\hat{x}^{(m)}\right)\right] - \mathbb{E}_{\hat{p}}\left[\hat{V}\left(\hat{x}^{(m)}\right)\right]\mathbb{E}_{\mathcal{P}^\pi}\left[\left[r\left(x^{(m)}\right) + \gamma V_{\text{tar}}\left(x^{(m+1)}\right)\right]\right]. \tag{50}$$

Combining this with Equation 44, we obtain

$$\mathbb{E}_{\hat{p}, \mathcal{P}^\pi} \left[ \hat{\mathcal{L}}_{m,b}^{\mathcal{P}^\pi}(\mathcal{P}^\pi, V | V_{\text{tar}}, x, x^{(m)}) \right] \tag{51}$$

$$= \mathbb{E}_{\hat{p}, \mathcal{P}^\pi} \left[ \left( \hat{V} \left( \hat{x}^{(m)} \right) - (\mathcal{T}_{\mathcal{P}^\pi} V_{\text{tar}}) \left( \hat{x}^{(m)} \right) \right)^2 \right] + \tag{52}$$

$$\mathbb{E}_{\hat{p}} \left[ \hat{V} \left( \hat{x}^{(m)} \right) \left[ \mathcal{T}_{\mathcal{P}^\pi} V_{\text{tar}} \right] \left( \hat{x}^{(m)} \right) \right] - \mathbb{E}_{\hat{p}} \left[ \hat{V} \left( \hat{x}^{(m)} \right) \right] \mathbb{E}_{\mathcal{P}^\pi} \left[ \left[ r \left( x^{(m)} \right) + \gamma V_{\text{tar}} \left( x^{(m+1)} \right) \right] \right]. \tag{53}$$

The first summand is the Bellman residual, for which the only minimizer is $\mathcal{T}_{\mathcal{P}^\pi} V_{\text{tar}}$. However, by Lemma 4, we can construct a function class so that $\text{Arg} \min \mathbb{E}_{\mathcal{P}^\pi} \left[ \hat{\mathcal{L}}_{m,b}^{\mathcal{P}^\pi}(\mathcal{P}^\pi, V | V_{\text{tar}}, x, x^{(m)}) \right] \not\subseteq \{ \mathcal{T}_{\mathcal{P}^\pi} V_{\text{tar}} \}$    □

While our proof only discusses the case $b = 1$, the same issue also appears with larger $b$. In many cases, the problem will be exacerbated by longer rollout horizons $m$ and $b$, as the variance of all functions involved grows with the time horizon.

### A.3. Main propositions: Section 6

#### A.3.1. PROPOSITIONS FROM BERTSEKAS & SHREVE (1978)

For convenience, we quote some results from Bertsekas & Shreve (1978). These are used in the proof of Lemma 5. While some of the definitions are rather technical, it is mostly sufficient to see a *stochastic kernel* as the continuous generalization of the transition matrix in finite MDPs. The projection used in Proposition 2 is simply a restriction of a set of tuples $(x, y)$ on the $x$ values. Other topological statements are standard and can be found in textbooks such as Munkres (2018). In the following $\mathcal{C}$ refers to sets of continuous functions.

**Proposition 1** (Proposition 7.30). *Let $\mathcal{X}$ and $\mathcal{Y}$ be separable metrizable spaces and let $q(\mathrm{d}y|x)$ be a continuous stochastic kernel on $\mathcal{Y}$ given $\mathcal{X}$. If $f \in \mathcal{C}(\mathcal{X} \times \mathcal{Y})$, the function $\lambda : \mathcal{X} \to \mathbb{R}$ defined by*

$$\lambda(x) = \int f(x, y) q(\mathrm{d}y|x)$$

*is continuous.*

**Proposition 2** (Proposition 7.33). *Let $\mathcal{X}$ be a metrizable space, $\mathcal{Y}$ a compact metrizable space, $\mathcal{D}$ a closed subset of $\mathcal{X} \times \mathcal{Y}$, $\mathcal{D}_x = \{y | (x, y) \in \mathcal{D}\}$, and let $f : \mathcal{D} \to \mathbb{R}^*$ be lower semicontinuous. Let $f^* : proj_{\mathcal{X}}(\mathcal{D}) \to \mathbb{R}^*$ be given by*

$$f^*(x) = \min_{y \in \mathcal{D}_x} f(x, y).$$

*Then $proj_{\mathcal{X}}(\mathcal{D})$ is closed in $\mathcal{X}$, $f^*$ is lower semicontinuous, and there exists a Borel-measurable function $\varphi : proj_{\mathcal{X}}(\mathcal{D}) \to \mathcal{Y}$ such that $range(\varphi) \subset \mathcal{D}$ and*

$$f [x, \varphi(x)] = f^*(x), \quad \forall x \in proj_{\mathcal{X}}(\mathcal{D}).$$

In our proof, we construct $f^*$ as the minimum of an IterVAML style loss and equate $\varphi$ with the function we call $f$ in our proof. The change in notation is chosen to reflect the modern notation in MBRL – in the textbook, older notation is used.

#### A.3.2. HELPER LEMMA

The first proposition relies on the existence of a deterministic mapping, which we prove here as a lemma.

**Lemma 5** (Deterministic Representation Lemma). *Let $\mathcal{X}$ be a compact, connected, metrizable space. Let $p$ be a continuous kernel from $\mathcal{X}$ to probability measures over $\mathcal{X}$. Let $\mathcal{Z}$ be a metrizable space. Consider a bijective mapping $\varphi : \mathcal{X} \to \mathcal{Z}$ and any $V : \mathcal{Z} \to \mathbb{R}$. Assume that they are both continuous. Denote $V_{\mathcal{X}} = V \circ \varphi$.*

*Then there exists a measurable function $f^* : \mathcal{Z} \to \mathcal{Z}$ such that we have $V(f^*(\varphi(x))) = \mathbb{E}_p [V_{\mathcal{X}}(x') | x]$ for all $x \in \mathcal{X}$.*

*Proof.* Since $\varphi$ is a bijective continuous function over a compact space and maps to a Hausdorff space ($\mathcal{Z}$ is metrizable, which implies Hausdorff), it is a homeomorphism. The image of $\mathcal{X}$ under $\varphi$, $\mathcal{Z}_{\mathcal{X}}$ is then connected and compact. Since $\mathcal{X}$ is metrizable and compact and $\varphi$ is a homeomorphism, $\mathcal{Z}_{\mathcal{X}}$ is metrizable and compact. Let $\theta_{V, \mathcal{X}}(x) = \mathbb{E}_{x' \sim p(\cdot|x)} [V(x')]$.

Then, $\theta_{V,\mathcal{X}}$ is continuous (Proposition 1). Define $\theta_{V,\mathcal{X}} = \theta_{V,\mathcal{Z}} \circ \varphi$. Since $\varphi$ is a homeomorphism, $\varphi^{-1}$ is continuous. The function $\theta_{V,\mathcal{Z}}$ can be represented as a composition of continuous functions $\theta_{V,\mathcal{Z}} = \theta_{V,\mathcal{X}} \circ \varphi^{-1}$ and is therefore continuous.

As $\mathcal{Z}_{\mathcal{X}}$ is compact, the continuous function $V$ takes a maximum and minimum over the set $\mathcal{Z}_{\mathcal{X}}$. This follows from the compactness of $\mathcal{Z}_{\mathcal{X}}$ and the extreme value theorem. Furthermore $V_{\min} \leq \theta_{V,\mathcal{Z}}(z) \leq V_{\max}$ for every $z \in \mathcal{Z}_{\mathcal{X}}$. By the intermediate value theorem over compact, connected spaces, and the continuity of $V$, for every value $V_{\min} \leq v \leq V_{\max}$, there exists a $z \in \mathcal{Z}_{\mathcal{X}}$ so that $V(z) = v$.

Let $h : \mathcal{Z}_{\mathcal{X}} \times \mathcal{Z}_{\mathcal{X}} \to \mathbb{R}$ be the function $h(z, z') = |\theta_{V,\mathcal{Z}}(z) - V(z')|^2$. As $h$ is a composition of continuous functions, it is itself continuous. Let $h^*(z) = \min_{z' \in \mathcal{Z}_{\mathcal{X}}} h(z, z')$. For any $z \in \mathcal{Z}_{\mathcal{X}}$, by the intermediate value argument, there exist $z'$ such that $V(z') = v$. Therefore $h^*(z)$ can be minimized perfectly for all $z \in \mathcal{Z}_{\mathcal{X}}$.

Since $\mathcal{Z}_{\mathcal{X}}$ is compact, $h$ is defined over a compact subset of $\mathcal{Z}$. By Proposition 2, there exists a measurable function $f^*(z)$ so that $\min_{z'} h(z, z') = h(z, f^*(z)) = 0$. Therefore, the function $f^*$ has the property that $V(f^*(z)) = \mathbb{E}_p[V(z')|z]$, as this minimizes the function $h$.

Now consider any $x \in \mathcal{X}$ and its corresponding $z = \varphi(x)$. As $h(z, f^*(z)) = |\theta_{V,\mathcal{Z}}(z) - V(f^*(z))|^2 = 0$ for any $z \in \mathcal{Z}_{\mathcal{X}}$, $V(f^*(\varphi(x))) = \theta_{v,\mathcal{Z}}(z) = \mathbb{E}_p[V_{\mathcal{X}}(x')|x]$ as desired.

$\square$

### A.3.3. MAIN RESULT: DETERMINISTIC MODEL

**Proposition 5.** *Let $\mathcal{X}$ be a compact, connected, metrizable space. Let $p$ be a continuous kernel from $\mathcal{X}$ to probability measures over $\mathcal{X}$. Let $\mathcal{Z}$ be a metrizable space. Consider a bijective latent mapping $\phi : \mathcal{X} \to \mathcal{Z}$ and any $V : \mathcal{Z} \to \mathbb{R}$. Assume that they are both continuous. Denote $V_{\mathcal{X}} = V \circ \phi$.*

*Then there exists a measurable function $f^* : \mathcal{Z} \to \mathcal{Z}$ such that we have $V(f^*(\phi(x))) = \mathbb{E}_p[V_{\mathcal{X}}(x^{(1)})]$ for all $x \in \mathcal{X}$.*

*Furthermore, the same $f^*$ is a minimizer of the expected IterVAML loss over any distribution $x \sim \rho$*

$$f^* \in \arg\min_{\hat{f}} \mathbb{E}_{\rho, \mathcal{P}^\pi}\left[\mathcal{L}_{\text{IterVAML},1}(\hat{f}, V_{\mathcal{X}}|V_{\mathcal{X}}, x)\right].$$

*Proof.* As we only deal with single steps here, we use $x^{(1)}$ and $x'$ interchangeably as the latter is simpler and more common nomenclature. The statement of the proposition itself is written with the $x^{(1)}$ notation to remain consistent with the main body of the paper.

The existence of $f^*$ follows under the stated assumptions (compact, connected, and metrizable state space, metrizable latent space, continuity of all involved functions) from Lemma 5.

First, expand the equation to obtain:

$$\mathbb{E}_{\mathcal{P}^\pi}\left[\hat{\mathcal{L}}_{\text{IterVAML},1}(f, V|V_{\mathcal{X}}x, x')\right] \tag{54}$$

$$= \mathbb{E}_{\mathcal{P}^\pi}\left[[V(f(\varphi(x))) - V_{\mathcal{X}}(x')]^2\right] \tag{55}$$

$$= \mathbb{E}_{\mathcal{P}^\pi}\left[\left(V(f(\varphi(x)))\right] - \mathbb{E}_{\mathcal{P}^\pi}\left[V_{\mathcal{X}}(x^{(1)})\right] + \mathbb{E}_{\mathcal{P}^\pi}\left[V_{\mathcal{X}}(x' - V_{\mathcal{X}}(x^{(n)}))\right)^2\right]. \tag{56}$$

After expanding the square, we obtain three terms:

$$\mathbb{E}_{\hat{p},\mathcal{P}^\pi}\left[\hat{\mathcal{L}}_{\text{IterVAML},1}(f, V|V_{\mathcal{X}}x, x')\right] = \mathbb{E}_{\hat{p},\mathcal{P}^\pi}\left[|V(f(\varphi(x))) - \mathbb{E}_{\mathcal{P}^\pi}[V_{\mathcal{X}}(x')]|^2\right] \tag{57}$$

$$+ 2\mathbb{E}_{\mathcal{P}^\pi}\left[[V(f(\varphi(x))) - \mathbb{E}_{\mathcal{P}^\pi}[V_{\mathcal{X}}(x')]]\left[\mathbb{E}_{\mathcal{P}^\pi}[V_{\mathcal{X}}(x')] - V_{\mathcal{X}}(x')\right]\right] \tag{58}$$

$$+ \mathbb{E}_{\mathcal{P}^\pi}\left[|\mathbb{E}_{\mathcal{P}^\pi}[V_{\mathcal{X}}(x')] - V_{\mathcal{X}}(x')|^2\right] \tag{59}$$

Apply the tower property to the inner term to obtain:

$$2\mathbb{E}_{\mathcal{P}^\pi}\left[\left[V\left(f\left(\varphi(x)\right)\right) - \mathbb{E}_{\mathcal{P}^\pi}\left[V_{\mathcal{X}}(x')\right]\right]\left[\mathbb{E}_{\mathcal{P}^\pi}\left[V_{\mathcal{X}}(x')\right] - V_{\mathcal{X}}(x')\right]\right] \tag{60}$$

$$= 2\mathbb{E}_{\mathcal{P}^\pi}\left[\left[V\left(f\left(\varphi(x)\right)\right) - \mathbb{E}\left[V_{\mathcal{X}}(x')\right]\right]\underbrace{\mathbb{E}_{\mathcal{P}^\pi}\left[\mathbb{E}_{\mathcal{P}^\pi}\left[V_{\mathcal{X}}(x')\right] - V_{\mathcal{X}}(x')|x'\right]}_{=0}\right] = 0. \tag{61}$$

Since the statement we are proving only applies to the minimum of the IterVAML loss, we will work with the $\arg\min$ of the loss function above. The resulting equation contains a term dependent on $f$ and one independent of $f$:

$$\arg\min_f \mathbb{E}_{\mathcal{P}^\pi}\left[\left|V\left(f\left(\varphi(x)\right)\right) - \mathbb{E}_{\mathcal{P}^\pi}\left[V_{\mathcal{X}}(x')\right]\right|^2\right] + \mathbb{E}_{\mathcal{P}^\pi}\left[\left|\mathbb{E}_{\mathcal{P}^\pi}\left[V_{\mathcal{X}}(x')\right] - V_{\mathcal{X}}(x')\right|^2\right] \tag{62}$$

$$= \arg\min_f \mathbb{E}_{\mathcal{P}^\pi}\left[\left|V\left(f\left(\varphi(x)\right)\right) - \mathbb{E}_{\mathcal{P}^\pi}\left[V_{\mathcal{X}}(x')\right]\right|^2\right]. \tag{63}$$

Finally, it is easy to notice that $V\left(f^*\left(\varphi(x)\right)\right) = \mathbb{E}_{\mathcal{P}^\pi}\left[V_{\mathcal{X}}(x')\right]$ by the definition of $f^*$. Therefore $f^*$ minimizes the final loss term and, due to that, the IterVAML loss.

$$\square$$

Comparing these results to Lemma 3, we can clarify one potential confusion about our results. The assumptions which are required for Proposition 5 ensure that the condition in Lemma 2 cannot be fulfilled in a continuous space, as there always exists a single point $y$ for which $f(y) = \mathbb{E}_p\left[f(x)\right]$. In this case, the optimal deterministic IterVAML model $f^*$ and the minimum variance distribution $p^*$ coincide. However, our definition of calibration requires the loss to recover optimal minima for any function class that includes them.

| Hyperparameter | Value | Hyperparameter (cont.) | Value (cont.) |
|---|---|---|---|
| Batch size | 128 | Model rollout depth $m$ | 1 |
| Discount $\gamma$ | 0.99 | Model bootstrap depth $b$ | Varied (0 and 1) |
| Actor learning rate $\alpha_\pi$ | 0.0003 | Model samples $k$ | Varied (1 and 4) |
| Critic learning rate $\alpha_Q$ | 0.0003 | Proportion real $\rho$ | 0.9 |
| Model learning rate $\alpha_{\hat{p}}$ | 0.0003 | Latent dimension | 512 |
| Encoder learning rate $\alpha_\varphi$ | 0.0001 | Gradient clipping | 10 |

*Table 2.* Hyperparameter of the deep learning experiments. Aside from the $(m, b)$-(C)VAML relevant hyperparameters, we keep all others consistent across environments and algorithms.

# B. Implementation Details – Section 5: Garnet

Each Garnet is defined by a randomly generated transition matrix and a random reward function. Across all experiments we use state spaces of size $|\mathcal{X}| = 50$. For each state $x_i$, we sample $k = 10$ successor states at random without replacement. Then we assign each successor state $x_j$ a weight $\omega_{ij} \sim \mathcal{N}(0, 1)$ and set the weight of all non-successor states to the floating point minimum. The transition matrix is computed via the softmax function. The reward is sampled as $r(x_i) \sim \mathcal{N}(0, 1)$.

This setup allows us to vary the stochasticity of the environment naturally. The softmax temperature parameter $\tau$ controls the stochasticity. As $\tau \to \infty$, the problem becomes deterministic with $p(x_j|x_i) = 1$ for the state $x_j$ with $\omega_{ij} = \arg\max_j \omega_{i,j}$. For $\tau \to \infty$, the probabilities for all successor states become equal $p(x_j|x_i) = \frac{1}{k}$. In our experiments, we vary $\tau$ such that for almost all Garnets we obtain full determinism and equal transition probabilities, which empirically happens in the range $\tau \in [0.001, 10.0]$.

# C. Implementation Details – Section 7: DM Control

## C.1. Architecture

The original MuZero paper (Schrittwieser et al., 2020) provides large-scale experiments on the ALE benchmark (Bellemare et al., 2013). However, the original algorithm and the ALE suite are computationally extremely expensive to run. In addition, to the best of our knowledge, no reliable open-source replication of the closed-source MuZero and Probabilistic MuZero models and algorithms exists. Instead, we use the TD-MPC architecture and algorithm (Hansen et al., 2022) as our main reference. This implementation uses the MuZero loss but is significantly cheaper and faster to run in a small-scale experimental setting, while still solving non-trivial control tasks.

We adopt the architecture and model-based MPC search procedure of Hansen et al. (2022) as our baseline model. In addition to the $L_2$ based deterministic latent setup, we also use a Gaussian latent model as our stochastic environment model. This implementation follows similar ideas used in observation-space models, such as Chua et al. (2018); Janner et al. (2019). While the stochastic environment models used in Dreamer (Hafner et al., 2021) would have been an alternative, it has not shown strong performance in continuous control tasks and is significantly more expensive to run. In addition, Voelcker et al. (2024) shows that observation-space reconstruction can be a suboptimal auxiliary task for aligning a latent space with value function prediction. Finally, we adopt the joint model-free and model-based training proposed in Voelcker et al. (2025). For the MuZero setting ($b = 1$) we update the value function both for the state and the predicted latent state. In the IterVAML setting, there is no meaningful way to update the model's predicted value function, and therefore we only update the current state's value function.

Code is provided in `https://github.com/adaptive-agents-lab/CVAML`.

Hyperparameters can be found in Table 2.

## C.2. Environment

As several recent papers have shown, the vast majority of environments in the DM Control suite are effectively saturated. We, therefore, restrict our experiments to the most difficult domains, *humanoid* and *dog*, on which large performance differences still appear (Nauman et al., 2024; Voelcker et al., 2025; Fujimoto et al., 2025).

## C.3. Additional results

In addition to the raw performance graphs in the main paper, we present the average model entropy and VAML loss curves in Figure 5. As expected, we find that the CVAML loss leads to a higher entropy model compared to the VAML loss. We compute the empirical differential entropy of the model prediction over the replay buffer as

$$- \sum_{x,a \in \mathcal{D}} p(z'|\varphi(x),a) \log p(z'|\varphi(x),a),$$

with $z'$ sampled from the latent model. This however does not lead to a noticeable difference in the VAML error itself. We also present the individual reward curves for each task in both domains in Figure 6.

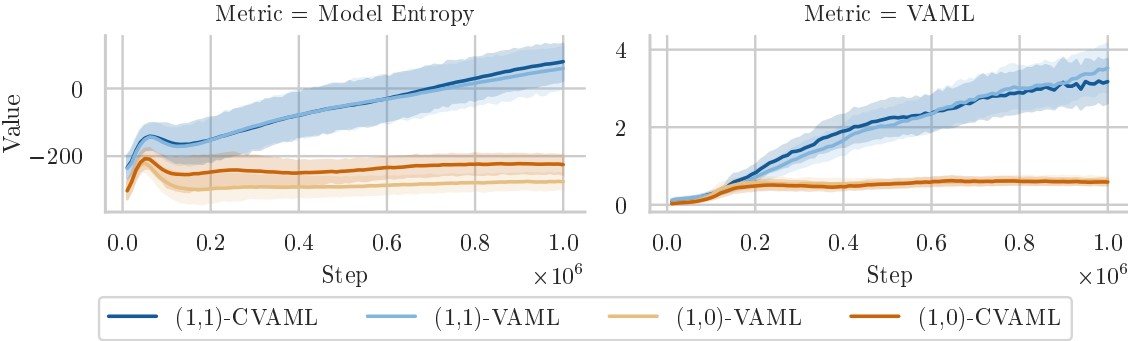

*Figure 5.* Model entropy (left) and loss value for the VAML-losses (right) aggregated across all environments. The model entropy of $(1, 0)$-CVAML differs significantly. However, this does not translate to a pronounced difference in the VAML error itself. Therefore, while the calibration term does lead to a higher variance, this does not necessarily translate into an improvement in performance. In $(1, 1)$-VAML compared with the calibrated version, the model entropy does not differ significantly. However, without calibration, the VAML error seems to be growing slightly stronger. Given the large deviation in performance we observe in the main results for $(1, 1)$-VAML, we conjecture that correcting the value function learning term is significantly more important for performance.

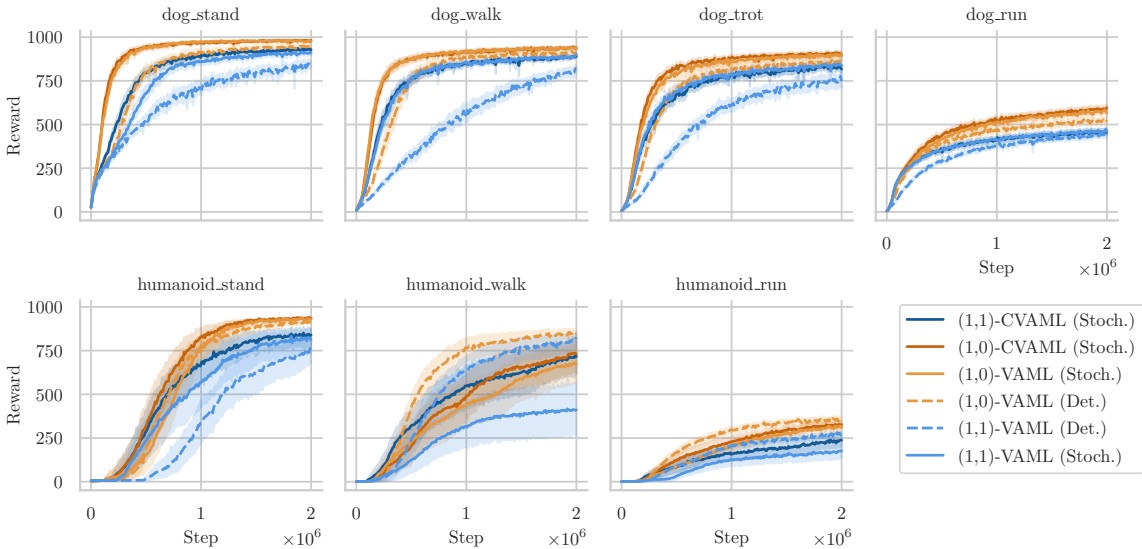

*Figure 6.* Episode returns per environment averaged over 30 seeds.

## C.4. Ablations

We provide additional ablations on the model loss here.

In the first set of experiments (Figure 7), we remove the CVAML loss from the model. We observe deteriorating agent performance, especially with the stochastic models, and in the humanoid tasks. In addition, we compare against a model-free TD3 baseline that uses the same architecture as our model based versions.

In the second set (Figure 8), we ablate the auxiliary loss. Again, we find that the auxiliary loss helps especially in the humanoid experiments. In addition, the $(1, 1)$-CVAML baseline suffers significantly more from the removal of the auxiliary loss than the $(1, 0)$-CVAML loss. Upon further investigation we observe that the bootstrapped TD target of the $(1, 1)$-CVAML loss leads to less stable learning compared to the value function target.

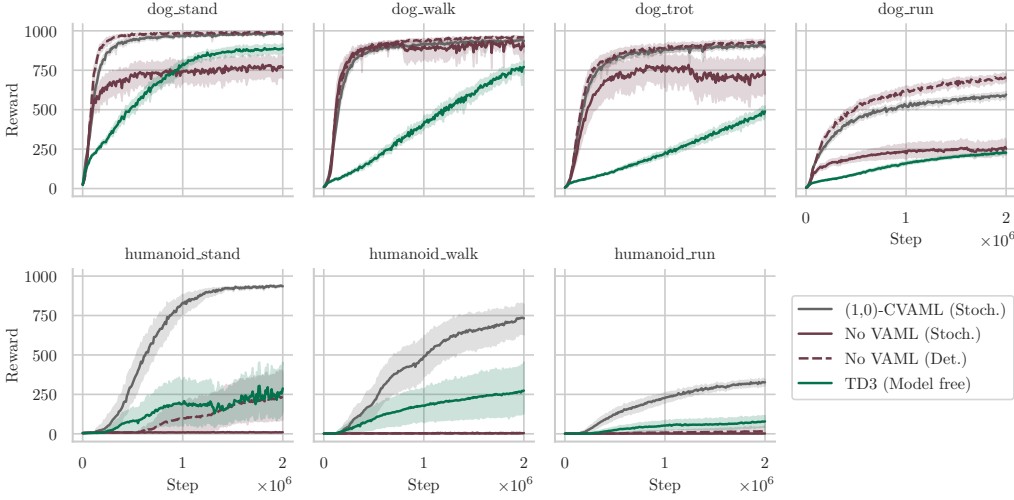

Figure 7. Performance of a model-free baseline (Fujimoto et al., 2018a), and the model-based algorithm using only the auxiliary loss for model training (No VAML), compared to (1,0)-CVAML. The model-free baseline fails to achieve strong returns on any problem. The No VAML baseline achieves slightly better returns on the dog_run task, but is unable to achieve any performance in the humanoid tasks.

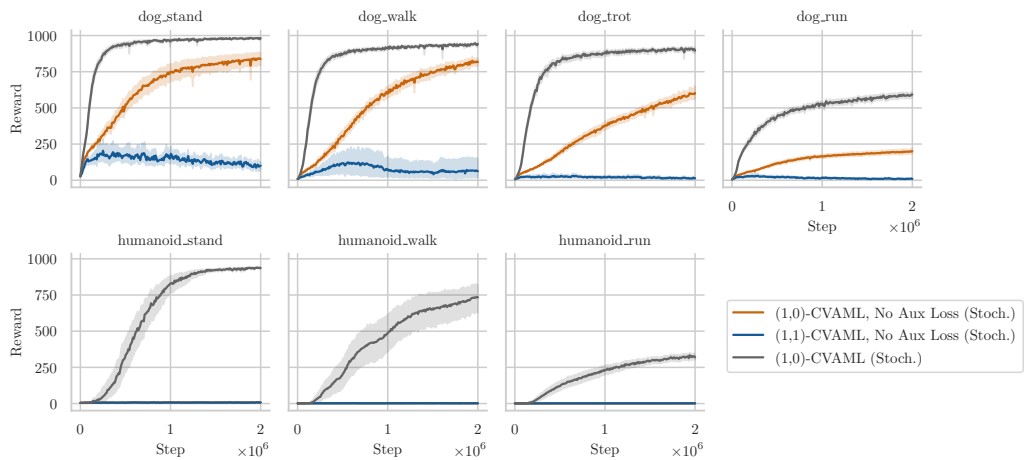

Figure 8. Performance comparison between agents trained using the $(1, 0)$-CVAML and $(1, 1)$-CVAML without the auxiliary loss to $(1, 0)$-CVAML trained with the auxiliary BYOL-style loss. The inclusion of the auxiliary loss is crucial for performance, both $(1, 0)$-CVAML and $(1, 1)$-CVAML without the auxiliary struggle on all the dog tasks, compared to the agent trained with both losses, the effect is even more pronounced for the $(1, 1)$-CVAML variant. On the humanoid task both variants without the auxiliary loss exhibit a complete failure, which further exemplifies the importance of this loss.

## C.5. Garnet Policy Iteration

Here we extend the Garnet experiments in Section 5 to the control case. In each iteration, the policy's value function is estimated with the model-based losses, following the Garnet experiments in the main paper. The policy is improved in a greedy fashion. The problem is a modified 5x5 cliffwalk environment https://gymnasium.farama.org/environments/toy_text/cliff_walking/ where the likelihood of moving in the intended direction is given by *temp*. Results are presented in Figure 9

Results are consistent with the garnet experiments in the main paper, except for the surprising outlier of the uncalibrated $(1, 1)$-VAML loss in the almost-deterministic case. This might be due to the different structures of the cliffwalk environment compared to the more diverse Garnet environments. However, this only holds for low rank deterministic case (top right corner).

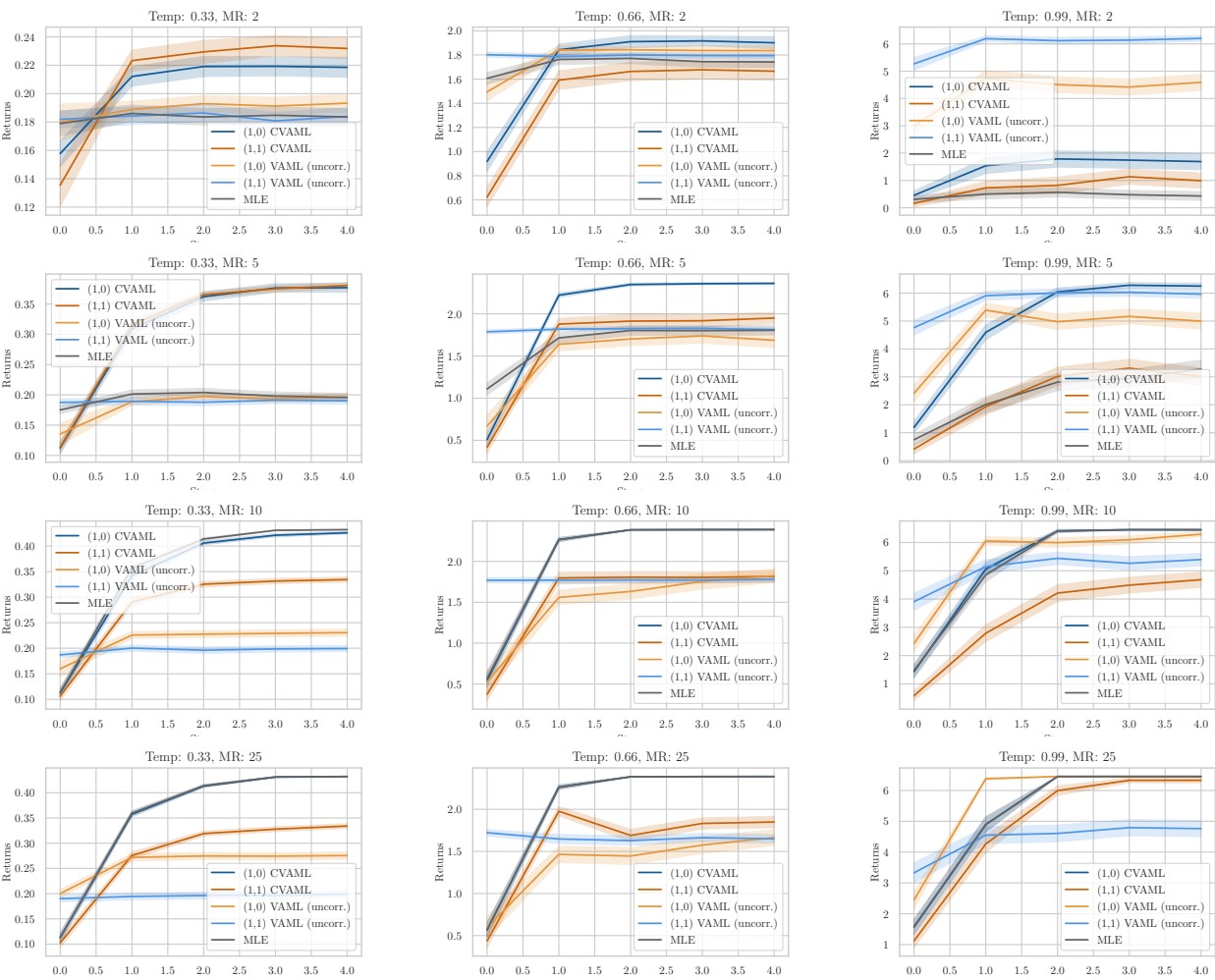

*Figure 9.* Return curves for cliffwalk policy iteration. Each curve shows confidence intervals over 1000 seeds. Each row is a different model rank (see description in the main paper), each column a different temperature.

