# OpenReview forum: "Calibrated Value-Aware Model Learning with Probabilistic Environment Models"
_ICML.cc/2025/Conference — ICML 2025 poster_

### Official Review · Reviewer_6JPt · 2025-03-12

**Overall Recommendation:** 3

**Summary:**

The paper investigates value-aware model learning (VAML), particularly examining the MuZero loss and Iterative VAML (IterVAML) within a unified framework termed (m, b)-VAML. The authors present theoretical insights, showing that standard (m, b)-VAML losses are generally uncalibrated surrogates when applied to stochastic models. The authors introduce a correction term leading to a calibrated version called CVAML to handle this. Additionally, the authors explore deterministic versus stochastic models, empirically demonstrating stochastic models' advantages in certain settings.

**Claims And Evidence:**

The claims are supported by evidence on a few environments: Garnet MDPs and two environments from the DMC suite. Although these results are insightful, a broader set of environments, possibly including more complex ones, would strengthen the claims.

**Essential References Not Discussed:**

I did not find any essential references that were not discussed.

**Experimental Designs Or Analyses:**

I have checked the soundness of the experimental designs and found no issues.

**Methods And Evaluation Criteria:**

The proposed method is simple yet makes sense for the given problem. As noted above, the provided evaluation criteria are not very thorough, but I think they are enough for proof-of-concept.

**Other Comments Or Suggestions:**

- The term sg is not defined until equation (2), even though it is first used in equation (1).
- In Table 1, the $m$ and $b$ conditions in the header columns seem to be shifted by one cell.

**Other Strengths And Weaknesses:**

Strengths
- The paper provides a nice unifying view of the value-aware model learning methods.
- The paper identifies a calibration issue that many value-aware model learning methods can suffer from.
- The proposed method is simple yet seems to address the issue effectively, given the empirical results.

Weaknesses
- Although the empirical experiments are insightful, they are somewhat limited to specific domains and scenarios. A broader set of environments or more extensive empirical validation might strengthen the claims about stochastic vs deterministic models.

**Questions For Authors:**

- L271: Is it correct that the indices $i$ and $j$ are inside the softmax, not outside?

**Relation To Broader Scientific Literature:**

Addressing calibration in surrogate losses for stochastic models is highly relevant to the reinforcement learning community, especially for those focusing on model-based approaches.

**Theoretical Claims:**

I have checked the theoretical claims' correctness and found no issues.

---

> ### Author Rebuttal · Authors · 2025-04-01
>
> Thank you for your review. We are thankful you took the time to engage with our paper! For your concerns on the number of environments and experiments, please refer to our reply to reviewer v9zD. Graphs can be found here https://drive.google.com/file/d/178cVcy05grmQ-dZCFu1p8ixIItgghoxG/view?usp=sharing

---

### Official Review · Reviewer_9s7Q · 2025-03-13

**Overall Recommendation:** 3

**Summary:**

The paper analytically investigates *value aware model learning* (VAML) and puts it into relation with the MuZero loss, by defining a generalizing (m,b)-VAML loss. The authors show that both approaches, that is, (1,0) and (1,1)-VAML, are *uncalibrated* because averaging over samples introduces a variance term, akin to Bellman Residual Minimization (BRM). However, in difference to BRM this variance term can be estimated with samples from the learned model, and the authors call the corresponding correction the *corrected VAML loss* (CVAML). CVAML outperforms VAML in a variety of studied cases of (m,b)-VAML on randomly generated toy MDP. Furthermore, the authors prove that (1,0)-VAML generally allows to learn deterministic transition models in latent spaces, even if the true model is stochastic. Classical auxiliary losses for latent space models introduce a bias for all but linear value functions, though. Experiments on two DMC environment demonstrate that using stochastic models can in some cases still be empirically beneficial, although the difference between VAML and CVAML auxiliary losses does not seem to be significant.

In summary, the paper discusses an important aspect of stochastic models in RL. It is generally well written and decently evaluated, but both points suffer in some parts which makes some contributions very hard to follow. I would recommend acceptance if some ambiguities in the formal notation and in the experimental description were rectified. I have not read the proofs in detail, though, so if other reviewers find faults, I would also be fine with rejecting the paper.

**Claims And Evidence:**

1. "Iterative VAML and MuZero value-aware model losses are not calibrated, but can be". This claim is shown analytically for simple cases of both losses and evaluated on many randomly generated toy MDP. The corrected Iterative VAML loss outperforms the corrected MuZero loss significantly, but only on unrealistic toy tasks.
2. "Deterministic (1,0)-VAML losses in latent spaces are sufficient to be value aware, even if the true dynamics are stochastic". This claim is analytically proven. The authors also find empirically that stochastic models can improve performance in one out of two DMC environments, but attribute this to induced robustness against deviations. Practical differences between CVAML and VAML seem to be insignificant, though.
3. "Auxiliary losses for latent dynamics models help learning, but introduce biases except for value functions that are linear in the latent space". This claim is mostly argued at an example. The empirical evidence only compares model-learning as an auxiliary task without clarifying wether the value function was linear in the last layer. The practical relevance of this claim is therefore questionable.

**Essential References Not Discussed:**

None that I know of.

**Experimental Designs Or Analyses:**

The empirical evaluation of the first claim is a bit underwhelming, as randomly generated MDP rarely have properties similar to real applications. Even an evaluation on gridworlds could improve the analysis. Nonetheless, the results are fairly clear and support the claim.

The evaluation of the second claim is a bit unclear: the authors train a model-free algorithm (TD3) with a VAML model-based auxiliary loss. This contradicts a bit the insight from their claim 3, unless the auxiliary loss is applied to the last layer of the value function (I could not find a mention thereof). Details are generally scarce here. MuZero would have made a much clearer test example, as it actually uses a value-equivalent model for inference. The results are also not very convincing, as the CVAML is not significantly better than VAML (except maybe for humanoid, but there the deterministic version is very strong). Please also add the results of vanilla TD3, as it is currently unclear whether VAML improves the performance at all.

**Methods And Evaluation Criteria:**

The claims are theoretically derived and empirically validated. Both parts can be improved by being more precise in formalism and details.

**Other Comments Or Suggestions:**

- l.61L: the MDP is missing an initial state distribution
- l.105L: it must be $\mathbb E_{\hat x^{(m)} \cdots}$
- Equation 2 is very (and IMHO unnecessarily) confusing. There are 8 parameters, but it seems you don't actually need $x^{(m)}$ (is always drawn from $\mathcal P^\pi$). Furthermore, this loss only makes sense if you define an expectation over it. How about this:
$\hat{\mathcal L}(m, b, p, p', V, V'|x) := \mathbb E[ (V(x^{(m)}) - [[T^b_{P^\pi} V'](x'^{(m)}))^2 | x \sim p, x' \sim p']$?
- Table 1: the (m,b) tuples in the second row have to shifted one column to the right
- The TD-loss in Equation 4 was very confusing to me. First, it should use $\hat V$, not $V$, as the value is learned. Second, an expectation over it (in the real model) still contains a variance term that cannot be reduced without double-sampling. I think you wanted to define $(\mathbb E_{\hat p}[V(x^{(m)})] - \mathbb E_{\mathcal P^\pi}[r^{(m)} + \gamma V_{tar}(x^{(m+1)})])^2$ to use in Proposition 4.
- Proposition 4 uses a surrogate loss with $\hat p = \mathcal P^\pi$, which does not contain a learned model $\hat p$! I assume you wanted to still use $\hat p$ of MuZero here. With this the (m,1) loss is indeed uncalibrated.
- Please refer to the appendix where the proofs can be found!
- I don't fully understand the issue exemplified in Equation 5. I would assume that the value loss is minimized for all $m \in \{0, \ldots\}$, so also for $m=0$. Your conclusion seems to be motivated by an empirical finding, not by this theoretical insight. Maybe there is another cause for it?
- Figure 2: I assume these are (1,0)-VAML losses, not (0,1) which would make no sense?
- Insight 1 mentions $\hat{\mathcal L}^{\hat p}_{0,j}$, but it is unclear what changing the superscript would do in Equation 2. Do you mean you learn the value by unrolling the model $\hat p$ for $j$ steps?
- l.265L: mention which distirbution the weights $\omega_{i,j}$ are drawn from
- l.248R: how is the value estimated? Analytically from the transition model?
- l.260R: why does (1,1)-VAML perform so poorly in deterministic environments?
- Proposition 5: It is not clear which loss $\hat{\mathcal L}_{IterVAML,1}$ refers to. The expected loss? The approximated? For which $k$?
- Equation 6: the (m,b)-VAML loss has a different signature than before. What does it mean specifically?
- l.363R: you mean (1,0)-updates!
- Please clarify that the performance improvement of the corrected loss (not calibrated loss) in Section 7 is **not** significant. A trend, not a reliable result!
- l.836, You are proving Proposition 4, not 2! Following propositions have the wrong numbers (except 5).

**Other Strengths And Weaknesses:**

The paper contains a small number of typos and inaccuracies that need to be fixed (see detailed comments). I also believe the formal notation can be massively improved, as sometimes the authors use losses which have not been explicitly defined, or where the general definition is ambiguous of how they should be read.

**Questions For Authors:**

- Why did you test latent CVAML on a model-free algorithm? Why not MuZero?

**Relation To Broader Scientific Literature:**

The paper is in parts very well written and discusses a wide range of relevant literature. The authors are commended for discussing many variants and differences to other methods in the main text.

**Theoretical Claims:**

The first theoretical claim is based on a simple, but elegant and to my knowledge unpublished, insight that learned stochastic models suffer from an additional variance term. The presented formalism, in particular the unified (m,b)-VAML loss (Equation 2), is very complex and in some parts unclear. I understand that the authors wanted to present a unified loss, but it is a bit unclear over which training sets the loss is defined, and how certain inputs and indices affect the loss (like in Proposition 4). The claim that it unifies the IterVAML loss is also quite questionable, as this is only true for $k=1$. However, the main claims on calibration (Proposition 2, 3 and 4) are interesting and correct (with some createve reformulations, please add links to the proofs in the appendix, though). Again, the definition of the TD loss in Equation 4 is hard to parse without any expectation over $x^{(m)}$.

The second theoretical claim (Proposition 5) is interesting, and I have not seen it stated like this before. It might be worth mentioning that bijective latent mapping are hard (or impossible) to come by. I did not follow the proof in detail here, though.

---

> ### Author Rebuttal · Authors · 2025-04-01
>
> Thank you for your kind and thorough review. We are grateful you took the time to engage very thoroughly with our paper! For your concerns on the number of environments and experiments, please refer to our reply to reviewer v9zD. These extended experiments also addresses concerns about significance. Graphs can be found here https://drive.google.com/file/d/178cVcy05grmQ-dZCFu1p8ixIItgghoxG/view?usp=sharing
>
> Notation: We share your concerns about the complexity of the notation. The loss has a lot of moving parts, and several past papers struggled to write it out properly in a non-confusing manner. We truly appreciate your suggestions for improvement and will adapt them where possible. As ICMl sadly doesn’t allow us to upload an updated PDF, we can only sketch out what we plan to change. All unmentioned typos will, of course, be corrected. Thank you for providing a thorough list.
>
> Equation 2: We are happy to simplify the equation as suggested!
>
> Equation 4: You are correct about $\hat{V}$. We will change that. The right-hand side is wrt to the target network and does not contain the model, so it does not have a double sampling issue. We will clarify this further by adding $[]_\mathrm{sg}$
>
> Equation 6: We dropped the dependency on $\hat{x}^{(m)}$ (as you suggested for Eq. 2) here already. We will unify this across the paper. We also split the model explicitly into an encoder and latent transition model, we’ll clarify this.
>
> Some concrete replies to confusing bits
>
> Proposition 4: We are analysing the value learning part of the MuZero loss here, so we simplified the setup by assuming that the learned model matches the ground-truth model. Our result specifically states that even if you assume that the model is perfect (equal to the ground-truth model), the MuZero value learning component will not learn the correct value function.
>
> Equation 5: We will clarify the writing here. Our concern is that the MuZero loss only enforces that the expectation of the next model state value function will match the bootstrapped target. Your insight is correct if each state x appears in the dataset as x^{(0)}. However, this might not be true due to the dataset being used. Then we cannot guarantee that model-generated states will have correct value estimates. This is not an empirical finding, simply a consequence of the loss formulation.
>
> Proposition 5: As we are looking at a deterministic model here, $k$ doesn’t change the loss (every sample would be the same), so we dropped it to declutter the presentation.
>
> Insight 1: Indeed, we will clarify the writing here. We meant a regular model-based target estimate with a j-step model rollout, similar to the one used in [1][2][3]
>
> L. 248: The ground truth for calculating the MSE is computed via the analytical solution over the transition matrix.
>
> l.260R: Why does (1,1)-VAML perform so poorly in deterministic environments?
>
> Note that while the environment is deterministic, the model is not. For the corrected (1,1) loss, we find that it is much more prone to get stuck in local minima due to the averaging effects of the MuZero style value update.
>
> On the large-scale experiments, we see a similar pattern even with deterministic models. We debated this question a bit among ourselves. Our conclusion is that the (1,1) loss updates the value function on a model-generated sample $\hat{x}^{(1)}$ while the (1,0) loss only updates the value functions on samples which actually appear in the replay buffer. As the model will have errors, this can cause additional difficulties with the loss function even on deterministic environment, as the sample $\hat{x}^{(1)}$ and target $T[V](x^{(1)}$ don’t match even if the model is not stochastic (simply due to model errors). Updating values on generated samples (not just with model-generated targets)
>
> Running TD3:
>
> This seems to be a misunderstanding. We will improve the writing, thanks for noticing! We did not run TD3, but we understand why this might appear to be the case. We ran TD-MPC1 with two modifications: we replaced SAC in TD-MPC1 with TD3 (which is likely where the confusion arises), and we used the model for value learning (similar to the way [1],[2],[3] use the model). This means the model here is used both for value improvement and also at inference time, as the MPC procedure uses the model and the value estimate for planning. We chose TD-MPC1 over a MuZero variant for a very simple reason: computational efficiency. MuZero is notoriously slow due to the overhead of MCTS and reanalyze, and few efficient open-source implementations exist. As our contribution is first and foremost a theoretical one, we decided to use the more lightweight TD-MPC algorithm, which still contains all the relevant parts of the MuZero algorithm (loss, model-based search, + added model-based value improvement).
>
> [1] MBPO https://openreview.net/forum?id=BJg8cHBxUS
>
> [2] Dreamer https://openreview.net/forum?id=S1lOTC4tDS
>
> [3] MAD-TD https://openreview.net/forum?id=6RtRsg8ZV1

---

### Official Review · Reviewer_Eb87 · 2025-03-16

**Overall Recommendation:** 3

**Summary:**

This paper examines a systematic issue in value-aware model learning (VAML) for reinforcement learning (RL).

Core Idea of VAML:
Unlike standard model learning, which maximizes log-likelihood, VAML optimizes for a model that results in zero value-function error. In other words, even if the model does not accurately capture the true transition dynamics of the MDP, it is considered "perfect" as long as it produces no value error.

Key Contribution of the Paper:
The paper highlights a critical issue with VAML-style objectives: when naively estimated empirically, they introduce additional bias. Specifically, the VAML objective requires an expectation over next states inside the squared error term, but empirical estimation places the expectation outside the square. This shift leads to overestimation due to Jensen’s inequality (since the squared function is convex). While the paper describes this as an "uncalibrated" estimate, a more standard term would be "biased."

Connection to Value Function Learning:
The paper makes a useful connection between this issue and value function learning in RL. A similar overestimation problem arises in residual learning, as described in Baird’s seminal work. Baird’s solution—using two next-state samples—mitigates this bias, though it is impractical in a model-free setting. However, in model-based RL, we can sample from the learned model and explicitly correct for the bias, effectively estimating and subtracting the overestimation error.

Practical Implications:
While the proposed bias mitigation approach is promising, its practical benefits remain uncertain. Reducing bias often comes at the cost of increased variance, and it is not always clear whether a zero-bias estimator is preferable to one with lower variance. The paper presents empirical evidence suggesting that this tradeoff is worthwhile in certain domains, but further theoretical and empirical validation is needed to confirm its general effectiveness.

**Claims And Evidence:**

The main claim—that VAML-style losses suffer from bias—is well-supported by both theoretical analysis and empirical results. The authors make a sharp observation about how the placement of the expectation affects estimation and introduce a clear argument grounded in Jensen’s inequality. This insight is particularly valuable because it connects VAML’s estimation issues to a well-known challenge in RL: overestimation bias in value function learning. The link to Baird’s residual learning framework strengthens the argument and provides a historical precedent for this type of error. While the paper does not explicitly frame the issue in terms of Jensen’s inequality, doing so makes the reasoning even more intuitive.

However, the secondary claim—that removing this bias in the prescribed manner is necessarily beneficial—is less well-supported. While the paper demonstrates that the proposed bias correction technique improves estimates in certain domains, the broader implications remain unclear. A key concern is the classic tradeoff between bias and variance: reducing bias can increase variance, sometimes to the detriment of overall performance. The authors acknowledge this tradeoff but do not thoroughly analyze how their correction method affects variance in different settings. While empirical results show an advantage in some cases, they do not provide a comprehensive theoretical justification or broader empirical validation across a wide range of environments.

Further, it is not always the case that a lower-bias estimator is preferable to one with slightly higher bias but significantly lower variance. Some RL methods explicitly tolerate bias in favor of stability and better long-term learning dynamics. A deeper analysis of this tradeoff—both theoretically and through more diverse empirical settings—would strengthen the paper’s argument for the practical benefits of its proposed correction.

**Essential References Not Discussed:**

NA

**Experimental Designs Or Analyses:**

My intuition about these VAML models is that while the point about likelihood models being an overkill is a fair one, in practice the correct VAML model might vary radically from one state to the other. This is because the correct next state to choose is the one with appropriate value function, and this can be a confounding factor. For example two very different next states have the same value function. Thus, when doing VAML in the function approximation case, generalization might become a big issue (two input states having radically different output).

Thus, I find it interesting that this paper is applying VAML to the latent space case where the issue explained above may be less severe.

On the negative side, a small portion of previous work in this space actually uses stochastic VAMLs. If the model is deterministic the bias issue is vanished, as identified by the paper. So while the bias issue is indeed present, it is not quite applicable in practice in light of the majority of papers using deterministic models anyways.


I think that the experimental results are quite limited. I would have loved to see the impact of this bias reduction idea in a much larger set of benchmarks. Other than the toy setting, the paper is evaluating the effectiveness on just two domains, which I feel is not sufficient to make a strong argument about the usefulness of the idea in practice. I still lean on the acceptance side in light of the nice observation and despite the weakness on the experimental side of things.

**Methods And Evaluation Criteria:**

Yes

**Other Comments Or Suggestions:**

NA

**Other Strengths And Weaknesses:**

NA

**Questions For Authors:**

I have an ongoing internal debate about the fundamental difference between VAML-based models and classical likelihood-based models, and I would love to hear the authors' perspective on it.

To me, VAML seems to introduce a kind of chicken-and-egg problem. The goal in VAML is to learn a model that minimizes value function approximation error when used for planning or learning. However, in the control setting, once the policy changes, the optimal model should also change. This is because the value function under the new policy might have different structure, meaning the previously learned model may no longer be optimal for minimizing value estimation error in the new setting. As a result, every policy update implies a corresponding update to the model, and every model update affects the policy, creating a tightly coupled iterative process.

This stands in contrast to classical likelihood-based models, where the objective is to approximate the true environment dynamics, which remain fixed regardless of policy updates. In this case, learning the model is a separate, more stable problem, and once the model is learned, it does not need to be modified with every policy update.

Given this difference, I’m curious how the authors view the stability and convergence properties of VAML-style models in iterative control settings. Does the need to repeatedly adapt the model with every policy change introduce instability or inefficiencies? Additionally, do the authors see scenarios where this iterative adaptation of the model and policy could be an advantage rather than a liability?

**Relation To Broader Scientific Literature:**

The connection to residual gradient method makes the result better situated.

**Theoretical Claims:**

I did check the proofs to the best of my ability. I think sometimes the notation is overly complicated. The main text could have presented things in the simplest case possible, and also not hold off to the main result on overestimation so much.

In equation (1), the sg notation is not immediately defined (deferred to the next page for some reason). It is also somewhat confusing, because on the term we are applying sg, there is no dependency to the model itself, so what does that even mean to say we are sg ing that term?

---

> ### Author Rebuttal · Authors · 2025-04-01
>
> Thank you for your kind and thorough review. We are grateful for your comments. For your concerns on the number of environments, please refer to our reply to reviewer v9zD. Graphs can be found here https://drive.google.com/file/d/178cVcy05grmQ-dZCFu1p8ixIItgghoxG/view?usp=sharing
>
> Bias-variance tradeoff: This is an excellent question and indeed the reason we chose to use the more technical term “calibration” instead of the term bias. We didn’t want to use “bias” as we felt it would be conflated too much with the phenomenon of bias and variance due to model architecture. However, in our case, we are talking about the bias of an estimator, not one stemming from a model class. As ML and statistical terminology are not always used in a consistent way across the literature, we felt that using the more specific idea of a calibrated surrogate loss would improve understanding.
>
> Our method does not induce a classic bias-variance tradeoff, as we can easily drive both terms to 0. Instead, we have a variance-compute tradeoff, as we can always draw more samples from our model to reduce the variance of the estimator, but that comes at the expense of additional computational steps. However, even for statistical estimators like ours, the unbiased estimator is indeed not guaranteed to be a minimum variance one for a fixed sample size. So there might be model classes and learning problems where the biased estimate does better with a fixed number of samples. However, as our method allows us to decrease the variance arbitrarily (with additional computational resources) we believe that it is still an important extension to the uncorrected estimator, which will result in non-zero error even in the limit of infinite model samples.
>
> Deterministic models: You are correct that the majority of published algorithms use a deterministic model. However, this is more of a historical issue than a reflection of the superiority of deterministic models. Most benchmark environments have a negligible amount of transition noise, and so using a stochastic model is often unnecessary. However, this is an artifact of the benchmarks the community has settled on, not really a condemnation of stochastic models.
>
> General concerns about VAML: The questions you raise are important and we believe unsettled in the literature so far. To the best of our knowledge, all current SOTA model-based RL approaches use some form of DAML loss [1][2][3][4]. Note that Dreamer [5] is an outlier, but seems to be outperformed in many cases by one of these approaches.
> However, constantly changing the target is indeed a concern. There is one interesting thing about the difference of (corrected) MuZero and IterVAML: For the true value function, the ground truth model is also a perfect model under the (corrected) VAML (MuZero and IterVAML) loss. However, for any other value function, this is still true for the IterVAML loss, but not for the MuZero loss. This might be an interesting insight as the VAML loss has at least one stable solution, while the MuZero loss does not enjoy this property.
>
> We added ablations to investigate your question further and were unable to achieve strong performance without the VAML or auxiliary components, so both seem important for these architectures.
>
> [1] Efficient MuZero https://openreview.net/forum?id=OKrNPg3xR3T
>
> [2] Efficient MuZero v2 https://arxiv.org/abs/2403.00564
>
> [2] TD-MPC1 https://www.nicklashansen.com/td-mpc/
>
> [3] TD-MPC2 https://openreview.net/forum?id=Oxh5CstDJU
>
> [4] MAD-TD https://openreview.net/forum?id=6RtRsg8ZV1

---

### Official Review · Reviewer_ABNg · 2025-03-19

**Overall Recommendation:** 3

**Summary:**

Value aware model learning losses penalizes if the model's value function prediction goes wrong. This works theoretically investigates the losses and shows that generally these losses are uncalibrated surrogate losses. They devise corrective measure for the losses. They provide experimental results on DMC control suite for their loss variant. They also show that learning a deterministic model can be sufficient but learning a calibrated stochastic model is more beneficial.

**Claims And Evidence:**

The claims are well supported by the theoretical evidence. They clearly poses the research questions and conduct relevant analysis.

**Essential References Not Discussed:**

N/A

**Experimental Designs Or Analyses:**

The experiments are relevant with the research questions. However, the experiments are limited to two DMC environment. To make the analysis robust, more experiments on diverse environments are crucial.

**Methods And Evaluation Criteria:**

As core of the Value-Aware Model Learning (VAML) framework, they consider the (m, b)-VAML family of losses and exhaustively consider the value categories for m and b. The analysis of variance and bias for stochastic model of the environment is crucial to understand loss performance. Further, the discussion on auxiliary losses makes the discussion comprehensive.

**Other Comments Or Suggestions:**

The paper should clearly mentions the definition of calibration early in the paper.

**Other Strengths And Weaknesses:**

The paper identifies key insights that would be very helpful to understand the relation between the model representation and loss functions. The proposed corrective loss components are of marginal novelty.

**Questions For Authors:**

N/A

**Relation To Broader Scientific Literature:**

The scope of the paper is a bit narrow. It considers a certain aspect of the model-based RL.

**Theoretical Claims:**

They claim that (m, b)-VAML family of losses are uncalibrated under a stochastic model. They successfully demonstrate that the error is dependent on the model’s stochasticity rather than the environment. They nicely carried out the analysis to show the advantage of the calibrated losses.

---

> ### Author Rebuttal · Authors · 2025-04-01
>
> Thank you for your review. For your concerns on the amount of environments, please refer to our reply to reviewer v9zD. Additional graphs can be found here https://drive.google.com/file/d/178cVcy05grmQ-dZCFu1p8ixIItgghoxG/view?usp=sharing
>
> Regarding your concern about novelty: while we agree that our focus is on model-based RL and therefore “narrow” in the sense that it targets a well-defined subarea of research, our results are, to the best of our knowledge, novel and unpublished. Model-based RL is also a mainstay topic at all major machine learning conferences, and we are therefore confident that this venue is an excellent place to disseminate our research.

---

### Official Review · Reviewer_v9zD · 2025-03-22

**Overall Recommendation:** 3

**Summary:**

The paper studies the family of value-aware model learning losses in model-based reinforcement learning, including MuZero loss. By theoretical analysis, it shows these losses are essentially uncalibrated surrogate losses. Then it proposes corrections for the losses. Experiments are conducted to show the correction to the losses is effective to obtain strong models.

**Claims And Evidence:**

The claims are well supported by evidence.

**Essential References Not Discussed:**

None.

**Experimental Designs Or Analyses:**

The experiment designs and analyses are generally sound. However, in Figure 4 the CVAML is not significantly better than VAML, which makes the claim less convincing.

**Methods And Evaluation Criteria:**

The proposed methods and the evaluation criteria make sense.

**Other Comments Or Suggestions:**

None.

**Other Strengths And Weaknesses:**

Strength
- The paper provides sound theoretical proof of the nature of losses used in the value-aware model learning, and proposes effective correction to them.
- The paper is well-written and easy to follow.

Weakness
- More experiments are needed to support the claims, e.g. about stochastic and deterministic models.

**Questions For Authors:**

None.

**Relation To Broader Scientific Literature:**

The paper is related to the losses used in value-aware model training.

**Theoretical Claims:**

Yes. The proofs for the theoretical claims are correct.

---

> ### Author Rebuttal · Authors · 2025-04-01
>
> Thank you for your review. We are using this reply as a general reply since all reviewers raised similar concerns and this review is the first one that appears on open review. As ICML does not allow an updated manuscript, but all reviewers are mostly concerned about empirical questions, we provide additional experiments under this URL https://drive.google.com/file/d/178cVcy05grmQ-dZCFu1p8ixIItgghoxG/view?usp=sharing
>
> ### Number of environments
>
> We acknowledge that our experiments cover fewer environments than other, more empirically focused papers. However, we see our main contribution in the theoretical aspects of our work. Therefore, our focus was not on providing a broad benchmarking comparison.
> We want to point out that we did not compare on 2, but on the 7 most challenging DMC tasks. We acknowledge that this was not stressed clearly in the paper and will update the writing accordingly. The graphs in Figure 4 show aggregated results over several tasks on each locomotion robot. We will provide individual curves in an updated document in the rebuttal URL. We would like to stress that this is already a broader comparison than e.g papers which compare on the standard Open AI Gym Mujoco environments [1,2,3], and the DMC dog and humanoid tasks are significantly more challenging.
>
> [1] CrossQ https://openreview.net/forum?id=PczQtTsTIX
>
> [2] MBPO https://openreview.net/forum?id=BJg8cHBxUS
>
> [3] ALM https://openreview.net/forum?id=MQcmfgRxf7a
>
> To provide additional evidence for our claims, we ran the following additional experiments:
>
> - We ran all the environments to 2,000,000 environment steps. This creates a clearer picture: the humanoid confidence intervals for (1,1)-VAML and CVAML do not overlap anymore.
> - We ran additional experiments as requested by the reviewers:
> - We ablated the TD-MPC1 model used by removing the auxiliary loss and the decision-aware loss respectively. We show that both variants decrease in performance.
> - We added a model-free TD3 baseline using the TD-MPC1 architecture (without model) for a full comparison. We show that this does not result in strong performance, especially on hard humanoid tasks.
>
> As requested by reviewer 9s7Q, we added a grid world experiment in which we conduct policy iteration.
>
> A general note regarding the scope of the empirical comparison: If the reviewers feel that specific environments would greatly aid in understanding the implications of our findings, we are happy to add them. We did so in the case of the gridworlds mentioned by reviewer 9s7Q. Otherwise, without concrete recommendations, it is hard to scope a proper reply to a general call for “more experiments”. Thus, we were wondering whether you had any particular environments in mind.

---

### Decision · Program_Chairs · 2025-05-01

**Decision:**

Accept (poster)

**Comment:**

This paper investigates value-aware model learning (VAML) for reinforcement learning (RL) and highlights a problem with the current method. Specifically, it argues that the family of (m,b)-VAML algorithms are uncalibrated when applied to a stochastic environment model. The authors propose a novel loss variant to address this issue and provide both theoretical and empirical evidence to support their solution.

All reviewers acknowledge that the main claim—that VAML-style losses are uncalibrated and that this issue can be corrected—is well-supported by both theoretical analysis and empirical results. However, some reviewers raised concerns that other claims in the paper were not sufficiently substantiated.

During the rebuttal period, the authors clarified ambiguities in the experimental section and provided additional experimental results. In summary, the authors’ responses adequately addressed the concerns raised by the reviewers.

In light of the reviewers’ assessments following the rebuttal, this paper is deemed suitable for acceptance.